



# The effects of decomposing invasive jellyfish on biogeochemical fluxes and microbial dynamics in an ultraoligotrophic sea

Tamar Guy-Haim[1], Maxim Rubin-Blum[1], Eyal Rahav[1], Natalia Belkin[1], Jacob Silverman[1], Guy Sisma-Ventura[1]

[1]Israel Oceanographic and Limnological Research, National Oceanography Institute, Haifa, 3108000, Israel

*Correspondence to*: Tamar Guy-Haim (tamar.guy-haim@ocean.org.il)

**Abstract.** Over the past several decades, jellyfish blooms have intensified spatially and temporally, affecting functions and services of ecosystems worldwide. At the demise of a bloom, an enormous amount of jellyfish biomass sinks to the seabed and decomposes. This process entails reciprocal microbial and biogeochemical changes, typically enriching the water column and seabed with large amounts of organic and inorganic nutrients. Jellyfish decomposition was hypothesized to be particularly important in nutrient-impoverished ecosystems, such as the Eastern Mediterranean Sea — one of the most oligotrophic marine regions in the world. Since the 1970s, this region is experiencing the proliferation of a notorious invasive scyphozoan jellyfish, *Rhopilema nomadica*. In this study, we estimated the short-term decomposition effects of *R. nomadica* on nutrient dynamics at the sediment-water interface. Our results show that the degradation of *R. nomadica* has led to increased oxygen demand and acidification of overlying water as well as high rates of dissolved organic nitrogen and phosphate production. These conditions favored heterotrophic microbial activity, bacterial biomass accumulation, and triggered a shift towards heterotrophic bio-degrading bacterial communities, whereas autotrophic pico-phytoplankton abundance was moderately affected or reduced. This shift may further decrease primary production in the water column of the Eastern Mediterranean Sea. Deoxygenation, acidification, nutrient enrichment and microbial community shifts at the sediment-water interface may have a detrimental impact on macrobenthic communities. Based on these findings we suggest that jelly-falls and their decay may facilitate an additional decline in ecosystem functions and services.

## 1 Introduction

Marine jellyfish often form massive aggregations, known as jellyfish blooms, with profound implications to human health, recreation and tourism, fisheries, aquaculture, and coastal installations (Purcell, 2012; Purcell et al., 2007; Richardson et al., 2009). Over the past three decades, a substantial increase in the frequency and intensity of jellyfish blooms has been documented worldwide (Attrill et al., 2007; Brotz et al., 2012; Licandro et al., 2010; Lynam et al., 2006; Quiñones et al., 2015; Shiganova et al., 2001) and was attributed to the growth in shipping, aquaculture and coastal protection (Duarte et al., 2013) or to natural global oscillations (Condon et al., 2013; Sanz-Martín et al., 2016). These blooms typically occur in 'boom and bust' cycles, where individuals suddenly appear in large numbers and shortly after disappear (Condon et al., 2013; Hamner



and Dawson, 2009; Schnedler-Meyer et al., 2018). This rapid collapse of jellyfish blooms *en masse* and their sinking to the seabed is a process commonly termed as 'jelly-falls' (Lebrato and Jones, 2011; Lebrato et al., 2012; Sweetman and Chapman, 2011).

During the blooms, jellyfish propagate by assimilating organic compounds of their prey, thus acting as a nutrient sink of organic carbon (C), nitrogen (N) and phosphorus (P) (Lebrato and Jones, 2011; Lucas et al., 2011; Pitt et al., 2009). The death

and sinking of jellyfish, followed by bacterial decomposition of their carcasses, lead to microbial community shifts (Kramar et al., 2019; Tinta et al., 2012; Titelman et al., 2006), resulting in oxygen depletion and acidification (Qu et al., 2015; Sweetman et al., 2016; West et al., 2008). On the seabed, jellyfish carcasses can be consumed by scavengers, thus acting as a rich carbon source that sustains benthic foodwebs (Hays et al., 2018; Sweetman et al., 2016; Sweetman et al., 2014). Both in the water column and on the sediment, jelly-falls undergo bacterial decomposition and play an important role in nutrient cycling (Chelsky

et al., 2016; Qu et al., 2015; West et al., 2008). The contribution of jellyfish degradation to nutrient cycling was hypothesized to be particularly important in nutrient-depleted, oligotrophic ecosystems (Pitt et al., 2009), such as the ultra-oligotrophic Eastern Mediterranean Sea (EMS), where microbial production is mainly limited by organic carbon (Sisma-Ventura and Rahav, 2019), nitrogen (Rahav et al., 2018b), or co-limited by nitrogen and phosphorus (Kress et al., 2005).

The most prominent jellyfish blooms in the Mediterranean Sea, particularly in its eastern basin, are caused by the scyphozoan

*Rhopilema nomadica* (Edelist et al., 2020; Katsanevakis et al., 2014) (Fig. 1). *R. nomadica* was first recorded in Israel in 1977 as a Lessepsian invader, introduced via the Suez Canal (Galil et al., 1990). Since then, it has expanded its distribution westwards with more frequent blooming occurrences (Balistreri et al., 2017; Edelist et al., 2020; Yahia et al., 2013). This species is venomous and its nematocysts contain active toxins, inflicting painful stinging on humans, as well as other adverse health problems, negatively affecting coastal recreation and tourism (Galil, 2018; Ghermandi et al., 2015). During blooms,

clogged intake pipes of power and desalination plants were reported in Israel (Angel et al., 2016; Galil, 2012). Reduced fishing harvests were also reported from Israel and Egypt, mostly due to net damage, loss of fishing days, and physical injury to the fishermen (Angel et al., 2016; Madkour et al., 2019; Nakar et al., 2011).

Although labeled as one of the worst invasive species in the Mediterranean Sea (Streftaris and Zenetos, 2006; Zenetos et al., 2010), the post-bloom decomposition dynamics of *R. nomadica* have never been investigated before. Here, we used incubation

experiments at the sediment-water interface to estimate the short-term decomposition effects of the invasive jellyfish *R. nomadica*, on (1) organic and inorganic nutrient dynamics and derived benthic fluxes, (2) bacterial abundance and production, and (3) microbial community composition, in the nutrient-impoverished EMS. We hypothesize that decomposed *R. nomadica* will trigger a rapid release of limiting nutrients, leading to enhanced fluxes to the sediment and overlying water, a substantial increase in bacterial abundance and production, and a shift in the microbial community composition and functions.



## 2 Methods and materials

### 2.1 Specimen collection and experimental setup

Three individuals of the scyphozoan jellyfish *Rhopilema nomadica* (Galil et al., 1990) of medium size (bell diameter 20-25 cm) were collected on the 29th of July 2019, at Tel-Shikmona, Haifa, near the Israel Oceanographic and Limnological Research Institute, on the shore of the easternmost Mediterranean Sea (Lat. 32°49'32"N, Lon. 34°57'26"E). The specimens were weighed and cut to pieces of 4-5 g to ensure representation of all body parts. Processed 25 g wet weight (ca. 1.25 g dry weight) of *R. nomadica* (including umbrellas, tentacles and oral arms, following Qu et al., 2015) were placed each in three Perspex cylinders (9.45 cm internal diameter; 50 cm length) that were filled up to 10 cm height with coastal sediments (Fig. 2), that were collected one week prior to the experiment, allowing the re-establishment of natural sediment profiles. Three additional cylinders with sediments did not include jellyfish and functioned as controls. The set up was completed by topping off the cylinders with oxygen saturated Mediterranean coastal water (ca. 3.14 L) pumped from 1 m depth and pre-filtered to remove large-size zooplankton (67 µm). The cores were sealed with gas tight sealing caps and placed in a lab with a relatively constant temperature of 27-28 °C, which is similar to the summer mean coastal water temperatures of the easternmost EMS (Raveh et al., 2015). The set up was acclimatized for 24 h to insure similar initial conditions in the chambers before jellyfish addition. Nutrient fluxes were measured using the whole core incubation technique previously described by Denis et al. (2001). Although restricting this study for testing short term responses, this method follows the best practices for measuring oxygen and nutrient fluxes and dynamics at the sediment-water interface is using sealed core incubations (Glud, 2008; Hammond et al., 2004; Pratihary et al., 2014; Skoog and Arias-Esquivel, 2009). Pre-filtered coastal water was transferred to a reserve tank, and stored under the same conditions as the incubated cores. The incubation cores were connected by tubing to the dedicated reserve tank, which replaced the water in the incubation chambers during each sampling.

Within each chamber, the overlying water was continuously mixed with a magnetic stirrer fixed 10 cm below the upper cap (75 rpm, Hammond et al., 2004), and were sampled at the following intervals: 0, 5, 10, 18, 26, 34, 44h, with dedicated sampling tubing. The reserve tank was sampled only at three intervals, 0, 20, 44h. At each sampling, 200 ml water samples were transferred to acid-washed transparent Nalgene bottles (250 ml), and sub-sampled by filtering (Minisart® 0.45 µm) for the following chemical analyses: $PO_4$, $NO_2+NO_3$ ($NO_x$), $Si(OH)_4$, $NH_4$, TDP (DOP), TN (DON). Nutrient samples were immediately frozen after collection for later analysis. Biological measurements were collected using unfiltered water for pico-phytoplankton (*Synechococcus*, *Prochlorococcus*, pico- and nano-eukaryotes), heterotrophic bacterial abundance and bacterial production measurements. Oxygen consumption rates at the sediment-water interface were continuously monitored using oxygen sensor spots (FireSting, PyroScience, Germany) adapted for measuring oxygen in closed containers through a transparent window (plastic or glass). The sensor spots were fixed to the inner side of the window with silicone glue. Four optical fibers continuously measured the oxygen in the three jellyfish chambers and one of the control incubations. The system was calibrated with saturated DIW. pH was measured with a sensor (MultiLine WTW, Germany) calibrated with NBS buffers.



Fluxes (mmol m$^{-2}$ d$^{-1}$) were determined by regressing the change in overlying water concentration (C) through time multiplied by the chamber height (Volume/Area), following Eq. (1):

$$f = \frac{dC}{dt} \times \frac{V}{A}$$

A correction for water replacement from the reserve tank was not applied, as the consequent error was less than 5%.

## 2.2 Inorganic and organic nutrients analysis

Nutrient concentrations were determined using a three-channel segmented flow auto-analyzer system (AA-3 Seal Analytical) following Kress et al. (2014). The limit of detection (LOD), measured as three times the standard deviation of 10 measurements of the blank (low nutrient seawater collected from the off-shore EMS), was 8 nM for $PO_4$, 50 nM for total dissolved phosphorus

(TDP) and $Si(OH)_4$, 80 nM for $NO_2+NO_3$ ($NO_x$) 90 nM for $NH_4$, and 0.74 µmol for total dissolved phosphorus (TDN). The accuracy of the analyses was determined using certified reference materials (CRM): MOOS 3 ($PO_4$, $NO_x$ and $Si(OH)_4$), VKI 4.1 ($NO_x$) and VKI 4.2 ($PO_4$ and $Si(OH)_4$). Results were accepted when measured CRMs were within ±5% of the certified values.

TDN and TDP were measured following potassium persulphate digestion and ultraviolet (UV) photo-oxidation, using a

digestion block system (Seal Analytical, UK). The reproducibility of the analyses was examined with VKI 4.2 and Deep Sea Reference (DSR) material. One of the TDP samples was lost (t= 44 h). DON concentrations were determined by subtracting $NO_x$ and $NH_4$ from TDN concentrations and DOP concentrations were determined by subtracting $PO_4$ from TDP concentrations.

## 2.3 Pico/nano -phytoplankton and heterotrophic bacterial abundance

Samples (1.8 ml) were fixed with flow-cytometry grade glutaraldehyde (0.02% final concentration, G7651, Sigma-Aldrich, USA), frozen in liquid nitrogen, and stored at -80 °C until analysis within two weeks. *Synechococcus* and *Prochlorococcus* , autotrophic pico/nano-eukaryotes (maximal size ~70 µm), and heterotrophic bacterial abundances were determined using an Attune® Acoustic Focusing Flow Cytometer (Applied Biosystems, USA) as described in Bar-Zeev and Rahav (2015). Samples of *Synechococcus*, *Prochlorococcus* and pico/nano-eukaryotes were run at 100 µL min$^{-1}$. Their taxonomic discrimination for

based on the orange fluorescence of phycoerythrin (585 nm), the red fluorescence of chlorophyll.*a* (630 nm), side-scatter (SSC, a proxy of cell volume), and on forward-scatter (FSC, a proxy of cell size.). Heterotrophic bacterial samples were run at 25 µL min$^{-1}$ using a discrimination threshold of green fluorescence (520 nm) and FSC. Beads (0.93 µm, Polysciences) were run in parallel as a size standard. Blank samples of sterile seawater (0.2 µm) were also run and their reads were removed from the total bacterial counts.



## 2.4 Bacterial production (BP)

Bacterial production was estimated using the $^3$H-leucine incorporation method (Perkin Elmer, specific activity 123 Ci mmol$^{-1}$) followed by micro-centrifugation (Simon, 1990). Samples (1.7 ml) were incubated with 10 nmol leucine L$^{-1}$ for 4-5 h under ambient temperature in the dark. Triplicate additions of trichloroacetic acid (TCA) were performed at each time-point and served as controls. The incubations were terminated with 100 µL of concentrated (100%) TCA. After adding 1 mL of scintillation cocktail (Ultima-Gold, PerkinElmer, USA) to each vial, the samples were counted using a TRI-CARB 2100 TR (Packard Biocience, USA) scintillation counter. A conversion factor of 3 kg C mol$^{-1}$ per every mole leucine incorporated was used, assuming an isotopic dilution of 2.0 (Simon and Azam, 1989).

## 2.5 DNA extraction and sequencing

Approximately 300 mL of overlying seawater were collected with a sterile syringe and passed through 0.22µm Sterivex filter. The membranes were removed from the cases, cut into pieces under sterile conditions and transferred into the extraction tubes. 250 mg from 0-1 and 1-2 cm sediment sections were transferred into the extraction tube. DNA was extracted from water and sediment using the DNeasy PowerSoil Kit (Qiagen, California, USA), using the manufacturer's protocol that included a FastPrep-24™ (MPBIO, Ohio, USA) bead-beating step (2x40 sec at 5.5 m/s, with a 5 min interval). The V4 region of the 16S rRNA gene was amplified using the modified primer pair 515F-806R (Apprill et al., 2015; Parada et al., 2016) in combination with CS1/CS2 tags (CS1_515Fc 5'-ACACTGACGACATGGTTCTACA GTGYCAGCMGCCGCGGTAA, CS2_806Rc 5'-TACGGTAGCAGAGACTTGGTCT GGACTACNVGGGTWTCTAAT), using the following PCR amplification protocol: initial denaturation at 94 °C for 45 s, 30 cycles of denaturation (94 °C for15 sec), annealing (15 cycles at 50 °C and 15 cycles at 60 °C for 20 sec) and extension (72 °C for 30 s). The 18S rRNA gene sequences were amplified using the 1391f-EukBr primer pair (Amaral-Zettler et al., 2009; Stoeck et al., 2010) in combination with CS1/CS2 tags (1391fc 5'-ACACTGACGACATGGTTCTACA GTACACACCGCCCGTC, EukBr 5'- TACGGTAGCAGAGACTTGGTCT TGATCCTTCTGCAGGTTCACCTAC), using the following PCR amplification protocol: initial denaturation at 94 °C for 45 s, 30 cycles of denaturation (94 °C for15 sec), annealing (60 °C for 20 sec) and extension (72 °C for 30 s). Library preparation from the PCR products and sequencing of 2x250 bp Illumina MiSeq reads was performed at HyLabs (Israel).

## 2.6 Statistical and bioinformatic analyses

Demultiplexed paired-end reads were processed in QIIME2 V2019.7 environment (Bolyen et al., 2018). Reads were truncated based on quality plots, checked for chimeras, merged and grouped into amplicon/environmental sequence variants (A/ESVs) with DADA2 (Callahan et al., 2016), as implemented in QIIME2. After removing the low-quality sequences, a total of 361335 (106169 in 6 and 255166 in 12 seawater and sediment samples, respectively) high-quality 16S rRNA gene amplicon reads with an average length of 260 bp, and a total of 658251 (162313 in 6 and 495938 in 12 seawater and sediment samples, respectively) high-quality 18S rRNA gene amplicon reads with an average length of 207 bp, were generated. The 16S and 18S



amplicons were classified with the Naïve-Bayes classifiers that were trained on the Silva 132 database, clustered at 99% (515F/806R region for the 16S and full-length sequences for the 18S rRNA gene amplicons). Downstream statistical analyses and plotting were performed in R (Core Team, 2020), using packages phyloseq (McMurdie and Holmes, 2013), ampvis2 (Andersen et al., 2018) and ggplot2 (Wickham, 2009). Systematic changes across experimental conditions were estimated with
DESeq2 (Love et al., 2014). The metabolic functions and pathways of the bacterial communities were predicted using Tax4Fun2 based on the KEGG database (Wemheuer et al., 2018). Pearson correlations and SIMPER analyses were performed in R using packages Hmisc (Harrell, 2004) and vegan (Oksanen et al., 2010). Principal component analysis of metabolic functions was performed with PAST V4 (Hammer et al., 2001).

## 3 Results

**3.1 Dissolved oxygen and pH dynamics**

Dissolved oxygen (DO) levels in the jellyfish treatments decreased from an initial average concentration of $261.5\pm4.5$ µmol·L$^{-1}$ to null within 40 hours, at an average rate of $5.9\pm0.1$ µmol·L$^{-1}$·h$^{-1}$, whereas the DO levels in the control chambers decreased slightly at an average rate of $0.7\pm0.1$ µmol·L$^{-1}$·h$^{-1}$ (mean ±SD, Fig. 3A). The variability within the treatment replicates and within the controls was small and non-significant (treatment replicates: $F_{(2,18)}=0.017$, $p=0.98$; controls: $F_{(2,18)}=0.055$, $p=0.59$).
The calculated average DO flux from the water column in the jellyfish treatment was $-56.9\pm1.0$ mmol m$^{-2}$ d$^{-1}$ versus $-6.7\pm0.3$ mmol m$^{-2}$ d$^{-1}$ in the controls (Table 1). In accordance with the decrease in DO, pH levels in the jellyfish treatments decreased from an initial average level of $8.10\pm0.02$ to $7.88\pm0.01$ and remained relatively stable (8.10-8.15) in the controls (Fig. 3B).

**3.2 Nutrient dynamics**

Nutrient levels significantly increased in the jellyfish-enriched chambers, whereas in the controls they remained stable and
low (Fig. 4). Ammonium was the dominant form of dissolved inorganic nitrogen in the experimental chambers. During the first 26 hours from the onset of the experiment, $NH_4$ levels increased at a rate of $0.27\pm0.12$ µmol·L$^{-1}$·h$^{-1}$, after which (26-44 h) the rate of $NH_4$ release increased to $1.33\pm0.31$ µmol·L$^{-1}$·h$^{-1}$ (Fig. 4A). $NO_2$ levels steadily increased at a rate of $5.5\cdot10^{-3}\pm2.0\cdot10^{-3}$ µmol·L$^{-1}$·h$^{-1}$, and decreased to background levels after 34 hours (Fig. 4B). $NO_3$ levels were generally higher in the jellyfish treatment than in the controls, but did not present any significant trend over time (Appendix A, Fig. A1). Silicic-acid
concentrations remained overall stable throughout the experiment, and higher in two of the jellyfish-enriched chambers (Appendix A, Fig. A1).

Within the first 5 hours following the jellyfish enrichment, orthophosphate levels increased by two orders of magnitude from $0.02\pm0.01$ to $1.02\pm0.13$ µmol·L$^{-1}$ (Fig. 4C). Throughout the rest of the experiment, $PO_4$ was fully consumed and its levels decreased to the background levels within 34 hours (0.04 µmol·L$^{-1}$), after which an increase was recorded (0.30 µmol·L$^{-1}$).
The majority of TDN and TDP released from the jellyfish was organic, where 84% of the TDN was DON, (Fig. 4D), and 71% of the TDP was DOP (Fig. 4E). Both organic nutrient levels significantly increased in the jellyfish enriched chambers, whereas





their concentrations in the control chambers remained stable and low. During the incubation period, DON concentrations increased 12-fold in the jellyfish treatment compared to the controls (Fig. 4D) and DOP concentrations increased 18-fold (Fig. 4E). The ratio between TDN and TDP (TDN:TDP) decreased from an initial average value of $96\pm18$ :1 to an average value of $23\pm7$ :1 in the jellyfish treatments, whereas in the controls it decreased to $57\pm3$ :1 (Fig. 4F).

The rates of nutrient release (remineralization rates) standardized to jellyfish biomass are detailed in Table 1, and the calculated nutrient fluxes (mmol m$^{-2}$ d$^{-1}$) in the jellyfish enriched cylinders and in the controls are summarized in Table2.

### 3.3 Autotrophic and heterotrophic abundance and bacterial production

Heterotrophic bacterial abundance increased linearly in the jellyfish treatments ($R^2$=0.98, p<0.01) and reached $1.5\cdot10^7 \pm1.9\cdot10^5$ cells·mL$^{-1}$ after 44 hours, whereas the controls remained stable at a concentration of $2.0\cdot10^6 \pm6.7\cdot10^4$ (Fig. 5A). *Synechococcus* abundance dropped in both jellyfish-enriched and control cylinders, however, after 44 hours, the number of *Synechococcus* cells in the jellyfish treatment was 5-fold larger compared to the controls (Fig. 5B). *Prochlorococcus* cell numbers increased in both jellyfish-enriched and control cylinders, and after 44 hours was lower in the jellyfish treatment (Fig. 5C). Both cell numbers of pico and nano -eukaryotes dropped throughout the experiment, nonetheless, were higher in the jellyfish treatment than in the controls by 50% (Figs. 5D-E).

Bacterial production remained stable in the jellyfish treatments at a rate of $3.1\pm0.3$ µg C ·L$^{-1}$ ·h$^{-1}$ during the first 26 incubation hours, increased to $4.3\pm0.1$ µg C ·L$^{-1}$ ·h$^{-1}$ and after 34 hours decreased again. Contrary, in the controls the bacterial production decreased immidiately from the onset of the experiment, and after 18 hours reached a rate of $0.4\pm0.2$ µg C ·L$^{-1}$ ·h$^{-1}$ that remained stable until the experiment ended (Fig. 5F).

The temporal dynamics of DO and nutrient concentrations strongly correlated with total bacterial abundance, but not with bacterial production (Appendix B, Table B1).

### 3.4 Microbial diversity

Bacterial alpha diversity (Fig. 6), was significantly lower in the jellyfish-enriched seawater than in the controls (p<0.05), but in the sediment samples there was no significant difference (p>0.05). The vast majority (93-97%) of the 18S sequence variants in seawater (Appendix C, Fig. C2) belonged to Scyphozoa, hindering alpha diversity evaluation. In the sediment, no significant difference (p>0.05) in alpha diversity was observed between treatments (Appendix C, Fig. C1). These findings were confirmed with rarefaction curves (Appendix C, Figs. C3, C4).

The distribution of the 30 most abundant bacterial genera measured in seawater in the jellyfish-enriched and control chambers is presented in a heatmap (as inferred from read abundance estimates, Fig. 7). Lineages for which significant changes in abundance (p<0.05) between the treatment and control were detected by DESeq2 (Fig. 7, yellow star symbols: lineages more abundant in the controls, purple star symbols: lineages more abundant in the jellyfish treatment). Nine lineages were significantly more abundant in the jellyfish treatment, whereas 12 lineages were significantly more abundant in the controls.



The relative abundance of the common marine bacteria, including the primary producers *Synechoccocus* and (chemo or photo)
the heterotrophic bacteria SAR11, HIMB11 and SAR86 (Dupont et al., 2012; Durham et al., 2014; Giovannoni, 2017), have
all diminished following jellyfish additions. Mostly opportunistic lineages (*Kordiimonadaceae*, *Pseudoalteromonadaceae*,
*Saccharospirillaceae* and *Nitrincolaceae*) that use multiple carbon sources, including xenobiotics, were enriched in jellyfish-
amended incubations, and are often associated with oil discharge (Yakimov et al., 2007). *Algicola* (*Pseudoalteromonadaceae*)
and *Kordiimonas* (*Kordiimonadaceae*) appear to be the most abundant degraders of the jellyfish biomass based on the marked
change observed in the abundance of their relative amplicon sequence variants.

Heatmap showing the distribution of the 30 most abundant genera in the sediment, measured in the 0-1 cm below surface layer
and in the 1-2 cm below surface layer (inferred from 16S sequences), in the jellyfish-enriched and control chambers is
presented in Fig. C1 (Appendix C). Among the 30 most abundant taxa, only *Fusimonas* and *Algicola* genera were significantly
more abundant in the jellyfish treatments in the 0-1 cm layer, however, in the 1-2 cm layer, there was no significant difference
between the treatments and controls.

The distribution of the 30 most abundant eukaryotic genera (inferred from the 18S rRNA amplicon read abundance) measured
in seawater and sediment in the jellyfish-enriched and control chambers is presented in Fig. C2 (Appendix C). Both sediment
layers showed no difference between treatment and controls, whereas in the seawater samples, four lineages of dinoflagellates,
*Ciliophora* and Labyrinthulomycetes were more abundant in the jellyfish than the controls.

Predicted functions were classified as KEGG orthologs (KOs) resulting in the identification of 346 KOs across all samples,
160 of which were associated with prokaryotic functions. The principal component analysis showed that jellyfish-treated and
control samples significantly differed based on microbial predicted functions (Fig. 8). Photosynthesis (ko00195) and carbon
fixation in photosynthetic organisms (ko00710) were enriched in controls, while catabolic functions, such as fatty acid
degradation (ko00071), valine, leucine and isoleucine degradation (ko00362) and xenobiotic degradation pathways, benzoate
degradation (ko00650) in particular were enriched in jellyfish additions (Fig. 8). SIMPER analysis (Appendix D, Table D2)
showed that the pathways mostly contributing to the difference between the jellyfish treatments and controls were signal
transduction 2-component system (ko02020) and ABC transporters (ko02010), contributing to 13% and 10% of the
dissimilarity between the groups, respectively.

## 4 Discussion

### 4.1 The effects of *R. nomadica* decomposition on oxygen and nutrient fluxes

Jellyfish blooms trigger substantial changes in dissolved oxygen, inorganic carbon and nutrient concentrations in the water
column (Condon et al., 2011; Pitt et al., 2009). Post-bloom processes, by comparison, modify the oxygen, carbon and nutrient
fluxes in the benthic boundary layer and the sediment-water interface (Chelsky et al., 2015; Lebrato and Jones, 2011; Qu et
al., 2015; West et al., 2008). Here we found that the decomposition of the invasive jellyfish *Rhopilema nomadica* triggered
deoxygenation of the seawater overlying the sediment to hypoxic and eventually anoxic levels. Similarly, increased sediment



oxygen demand following jellyfish decomposition was measured by West et al. (2008) in *Catostylus mosaicus* and by Tinta et al. (2016) in the moon jellyfish *Aurelia aurita*. Qu et al. (2015) that studied the effects of *Cyanea nozakii* decomposition in the Yellow Sea using incubations found that oxygen was depleted in both sediment and seawater. They hypothesized that the metabolism and propagation of heterotrophic bacteria led to enhanced oxygen consumption. Indeed, our experimental results

support this hypothesis, as bacterial abundance was strongly correlated with oxygen levels, whereas the abundance of autotrophic cyanobacteria decreased as they were likely outcompeted by the heterotrophic bacteria (Sisma-Ventura and Rahav, 2019; Thingstad et al., 2005). Thus, jelly-falls can generate hypoxic areas on the seabed and overlying waters (Pitt et al., 2009), and affect the benthic infauna (Chelsky et al., 2016). Although the Eastern Mediterranean coastal waters are well-oxygenated (Kress et al., 2014), the collapse of massive *R. nomadica* blooms could potentially create local hypoxic or even anoxic hotspots

on the seabed, thereby affecting the surrounding biota (Feely et al., 2010).

In addition to deoxygenation, our experiment showed a significant reduction in pH, to levels that are considered detrimental to various organisms, mainly calcifies (Kroeker et al., 2010; Zunino et al., 2017). Acidification as a result of jellyfish decomposition was also observed by Qu et al. (2015) that speculated that the release of amino-acids and fatty-acids from proteins and lipid metabolism of jellyfish tissue is the root cause for the observed decrease in pH. Nonetheless, hypoxia and

acidification are biogeochemically coupled via the production of inorganic carbon in the process of respiration (Feely et al., 2010; Gobler and Baumann, 2016). In addition, increase in $NH_4$, as was measured in our experiment, increases total alkalinity and pH, whereas nitrate and silicate decrease pH, but they were comparably scarce. Based on oxygen to carbon conversion (1:1.3), and alkalinity change due to $NH_4$ addition, it is estimated that the observed decrease in pH in our experiment can be solely attributed to inorganic carbon and carbonic acid production (due to bacterial respiration) and ammonium release. The

combination of hypoxia and acidification may have synergistic additive negative effects on the benthic fauna (Gobler et al., 2014; Melzner et al., 2013). Furthermore, ammonium in high concentrations may have toxic effects on various marine organisms, from bacteria to fish (Brun et al., 2002; Eddy, 2005; Ferretti and Calesso, 2011; Müller et al., 2006).

The decomposition of dead *R. nomadica* tissue generated an immediate rapid release of organic and inorganic phosphate after which the inorganic phosphate ($PO_4$) was completely consumed, while the efflux of organic and inorganic (mostly ammonium)

nitrogenous compounds gradually increased throughout the experiment. Similar dynamics were observed in *C. mosaicus* by West et al. (2008) and Chelsky et al. (2015), and by Tinta et al. (2010) in *Aurelia solida*, where organic and inorganic phosphate peaked and completely abolished within 24 hours, presumably due to bacterial uptake. The production of $NO_x$ in our experiment was evident only in the jellyfish treatment while oxygen levels were conducive, suggesting that nitrification plays an important role in nutrient dynamics following jellyfish decomposition, as was found in different jellyfish species (Hubot et

al., 2020; Welsh et al., 2009). The stoichiometric relationship between TDN and TDP decreased from 57:1 to 23:1 as a result of *R. nomadica* decomposition, as was also found by West et al. (2009) and Qu et al. (2015). Elemental body composition of scyphozoan jellyfish, in general, is 2.48 N %DW (dry weight) and 0.22 P %DW, hence an N:P ratio of 11:1 (Lucas et al., 2011). The higher N:P measured in our experiment indicates a mismatch between the resource (i.e., jellyfish organic matter) and consumers (e.g., bacteria). This elemental imbalance determines ecological interactions and metabolic rates (Sterner and





Elser, 2002). Thus, the higher N:P measured here may imply on a preferential bacterial retention of phosphate (West et al., 2008). A recent study from the same area showed that the addition of organic nutrients stimulated heterotrophic microbial biomass and activity (Sisma-Ventura and Rahav, 2019), highlighting the importance of nutrient remineralization in this ecosystem.

The rates of nutrient release from *R. nomadica* decomposition found in this study were comparable to jellyfish decomposition-

driven rates found in former studies (e.g., Blanchet et al., 2015; Pitt et al., 2009; Qu et al., 2015; Tinta et al., 2012; Tinta et al., 2016; Titelman et al., 2006; West et al., 2008). Ammonium release rate in *R. nomadica* (1.96 µmol $g^{-1}$ WW $d^{-1}$) was slightly higher than the rate measured by Tinta et al. (2012) in *Rhizostoma pulmo* (1.6 µmol $g^{-1}$ WW $d^{-1}$), another common Mediterranean scyphozoan. Reported densities of *R. nomadica* aggregations from the EMS are $1.6 \cdot 10^5$ $km^{-2}$ in the Israeli coast (Lotan et al., 1992; Lotan et al., 1994), $1 \cdot 10^6$ $km^{-2}$ in the Lebanese coast (Lakkis and Zeidane, 1991), and $9 \cdot 10^5$ $km^{-2}$ in the

Mediterranean Egyptian coast (Madkour et al., 2019). The average wet weight of *R. nomadica* changes seasonally, 1340 ±953 g $ind^{-1}$ during summer and 2450 ±1854 g $ind^{-1}$ during winter (N=40, T.G.-H. unpublished data), yielding ca. 1.3 kt $km^{-2}$. We can, therefore, estimate that the collapse of *R. nomadica* bloom potentially releases ammonium and phosphate in concentrations of 2.5 and 0.8 kmol $km^{-2}$, respectively.

Nutrient remineralization during jelly-fall decomposition, as was found in this study and others, can be inhibitory or toxic to

some organisms (e.g., dissolved sulfides and ammonium in Chelsky et al., 2016), but on the other hand, can stimulate primary production and induce algal blooms in the water column and on the sediment. Møller and Riisgård (2007) found that following blooms of *A. aurita*, peak concentrations of chlorophyll-*a* were measured in a heavily eutrophied Danish Fjord. Using mesocosm experiments, West et al. (2009) found that excretion of jellyfish *C. mosaicus* led to a 10-fold increase in diatom abundance. In the EMS, *R. nomadica* typically peaks in the summer months and collapses at the end of July (Edelist et al.,

2020), whereas peak chlorophyll-*a* concentrations in the water column are measured during wintertime (Ignatiades et al., 2009; Rahav et al., 2018a; Raveh et al., 2015). This may result from the competitive exclusion of phytoplankton by heterotrophic bacteria (Sisma-Ventura and Rahav, 2019). Thus, fertilization of the water column due to nutrient release from *R. nomadica* decomposition may fail to trigger an algal bloom in the EMS. In contrast, maximum chlorophyll concentrations were measured in the sediment of the shallow Israeli coastal shelf during the late spring-summer (Hyams-Kaphzan et al., 2009; Tadir et al.,

2017). This discrepancy was explained by the spring bloom of benthic primary producers. However, the results of this study could provide another plausible explanation for the high summer chlorophyll concentrations in the sediment, which may be the post-bloom nutrient boost to the benthic ecosystem.

## 4.2 Decomposition induced shifts in bacterial community abundance, production, composition and functionality

Heterotrophic bacteria are major consumers of dissolved organic matter (DOM) in marine ecosystems and can therefore benefit

from jellyfish decomposition. Previous studies have demonstrated a significant increase in bacterial abundance triggered by jellyfish degradation (Blanchet et al., 2015; Condon et al., 2011; Dinasquet et al., 2012; Frost et al., 2012; Kramar et al., 2019; Tinta et al., 2016; Tinta et al., 2010; Titelman et al., 2006; West et al., 2009). Our study found that the decomposition of *R.*





*nomadica* induced an increase in two orders of magnitude in the heterotrophic bacteria abundance. Autotrophic cyanobacteria, on the other hand, decreased (*Prochlorococcus*), or increased to a lower level than the control (*Synechococcus*), likely due to

deoxygenation (Bagby and Chisholm, 2015) or out-competition by heterotrophic bacteria (Sisma-Ventura and Rahav, 2019; Thingstad et al., 2005).

The fate of jellyfish DOM consumed by bacteria depends on bacterial growth efficiency—the ratio of bacterial production to substrate assimilation (i.e., the sum of bacterial production and respiration) (Condon et al., 2011). While some studies have found that the succession of bacterial production mirrored bacterial abundance and respiration (Blanchet et al., 2015; Titelman

et al., 2006), in our study, bacterial production reduced in the controls, whereas under jellyfish enrichment remained at a steady, eightfold higher, level. This decoupling between bacterial abundance and production may indicate a shift in the functional diversity and metabolic demands of the jellyfish-associated bacterial communities along the experiment. In the shallow coastal waters of the EMS, bacterial production levels peak in winter and in summer (Raveh et al., 2015), coinciding with, and potentially contributed by, the seasonal aggregations of *R. nomadica* (Edelist et al., 2020).

A significant reduction in the microbial α-diversity indices of seawater during jellyfish decomposition was observed in this as well as in former studies (Blanchet et al., 2015; Kramar et al., 2019; Tinta et al., 2012). The decline in diversity can be attributed to the specialization of surface-colonizing bacteria, having the competitive advantage for settling from the surrounding seawater (Kramar et al., 2019), and was thus less evident in the sediment samples. Additionally, changes in bacterial diversity may result from bacterial antagonism, i.e. the production of antagonistic compounds and sensitivity or resilience to them

(Titelman et al., 2006). In this study, we found a significant increase in the relative abundance of the Alphaproteobacterium *Kordiimonas* and the Gammaproteobacteria *Algicola* in the seawater enriched with *R. nomadica*. Similarly, the predominance of Alphaproteobacterium and Gammaproteobacteria stimulated by jellyfish decomposition was found in different studies (Basso et al., 2019; Blanchet et al., 2015; Condon et al., 2011; Dinasquet et al., 2012; Kramar et al., 2019; Tinta et al., 2012; Titelman et al., 2006). Gammaproteobacteria are conspicuous particle colonizers (Bižić-Ionescu et al., 2015; Simon et al.,

2002), capable of degrading high molecular weight organic compounds (Cottrell and Kirchman, 2000; Reichenbach, 1992; Woyke et al., 2009), e.g. hydrocarbons (Niepceron et al., 2013). Kramar et al. (2019) found that Alphaproteobacteria and Gammaproteobacteria dominated the body surface of *Aurelia*, especially during the senescent phase. Blanchet et al. (2015) found a succession of bacterial diversity during the degradation of *Aurelia* and concluded that Alphaproteobacteria and Gammaproteobacteria have a major role in the succession of jellyfish DOM degradation. The link between the bacterial

diversity of living *R. nomadica* at different life phases and the diversity of bacteria associated with its decomposed DOM is yet to be investigated.

Both genetic and functional diversity analyses of bacterial communities demonstrated a shift under *R. nomadica* degradation. We found that the predicted functions that dominated the decomposed jellyfish communities were signal transduction (2-component system), catabolic functions, such as fatty acid degradation, valine, leucine and isoleucine degradation, xenobiotic

degradation pathways, and benzoate degradation. In the control communities, predominating functions were photosynthesis and carbon fixation in photosynthetic organisms. This functional shift can be explained by the fact that autotrophic





cyanobacteria may be outcompeted by bio-degrading heterotrophic bacteria. Once the jellyfish bloom decomposes, populations of these intrinsic microbial bio-degraders become dominant and active, exploiting the carbon and nutrients released from the jellyfish. Using 16S rRNA amplicon data for predicting functional profiles is a powerful tool for assessing bacterial functional

diversity, nonetheless, its accuracy and resolution are dependent on the representation of sampled organisms in the 16S rRNA and KEGG databases (Sun et al., 2020; Wemheuer et al., 2018). Likely, jellyfish degraders are under-represented in these databases. Further research using omics (e.g., whole-genome sequencing) will elucidate the metabolic potential of microbial degraders of the jellyfish necromass.

Although not to the same extent as bacterial diversity, eukaryotic diversity had too, shifted during the decomposition of *R.*

*nomadica*, to a more flagellate-dominated community. Marine ciliates and parasitic protists (Labyrinthulomycetes) were also more abundant in the jellyfish decomposed community. Flagellate bacterivory represents the primary mechanism for the reintroduction of jellyfish carbon into the planktonic food web (Condon et al., 2011; Gasol and Kirchman, 2018). The increase in ciliates can be attributed to a "bottom-up" effect, where with the increase in flagellates, the abundance of their predators (e.g., ciliates) also increases. Since jellyfish consume ciliates (Kamiyama, 2018; Stoecker et al., 1987), the flagellate carbon

could be assimilated and recycled by the jellyfish, creating a positive-feedback loop termed as the "jelly-loop" (Condon et al., 2011; Lebrato and Jones, 2011).

## 5 Conclusions

Our study examined, for the first time, the decomposition dynamics of the bloom-forming invasive jellyfish *R. nomadica* in the Mediterranean Sea. The geographical distribution of this venomous species is continuously expanding, and its outbreaks

are becoming more frequent, large, prolonged, with numerous negative impacts on human health, marine infrastructure, tourism, and fisheries.

We found that the jellyfish degradation had a significant influence on the fluxes of organic and inorganic nutrients at the sediment-water interface, transforming the microbial community composition and functions. The high rates of organic nitrogen and phosphate release favored heterotrophic-dominated metabolism, leading to a shift towards heterotrophic bio-degrading

bacterial communities. This shift may further decrease primary production under the ultra-oligotrophic regime of the Eastern Mediterranean Sea. On the seabed, hotspots of deoxygenated, acidified, and nutrient-rich sediment may alter microbial and macrobenthic communities.

Future investigations on the decomposition dynamics of *R. nomadica* should be conducted in larger experimental systems (i.e., mesocosms) or in-situ, under more realistic conditions. The effects of environmental change drivers, such as warming,

acidification, or anthropogenic pollution should also be tested. Additionally, the consumption of jelly-falls by scavengers in the Eastern Mediterranean Sea should be explored. This and future studies will shed light on the variable effects of the reoccurring massive blooms on the ecosystem functions and services in this rapidly changing environment.

**Data availability**

All data were deposited in an Open Access data archiving and publication repository (Pangaea, a member of the ICSU World
Data System) and are available at https://doi.pangaea.de/10.1594/PANGAEA.915464. Amplicon reads were submitted to
NCBI Sequence Read Archive BioProject PRJNA626084.

**Author contribution**

This work was conceived by all authors. TGH and GSV led the research and performed the experiments, MRB conducted the
microbial diversity and bioinformatic analyses, ER and NB analyzed the microbial abundance and production, JS contributed
to the study conception. TGH wrote the manuscript with substantial contributions from all co-authors. All authors have read
and approved the final submitted manuscript.

**Competing interests**

The authors declare that they have no conflict of interest.

**Acknowledgments**

We would like to thank Dina Kolker for helping with the nutrient analysis and Dar Golomb for preparing the incubation
cylinder illustration. This study was partially supported by the National Israeli monitoring program.

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





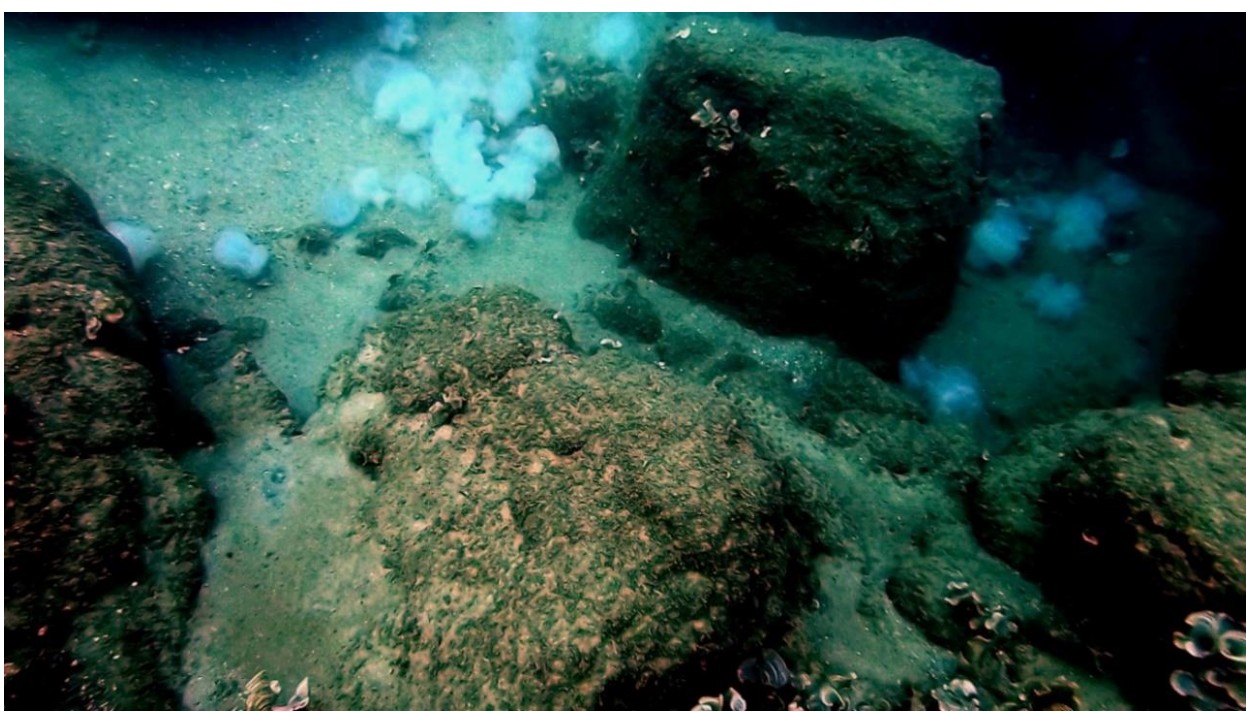

**Figure 1. Jelly-falls (carcasses) of ca. 30 *Rhopilema nomadica* in the Mediterranean coast of Caesarea, Israel. 8-9 m**
**depth, photographed on 27 July 2019 after the typical peak summer bloom (Photo: Zvika Fayer).**

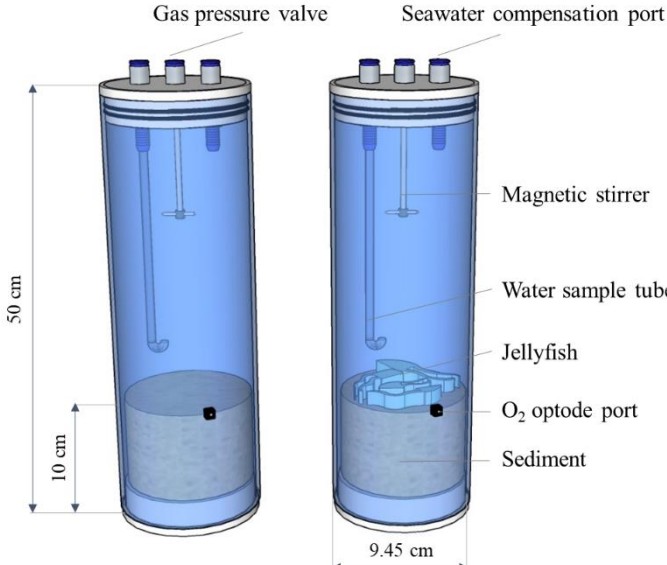

**Figure 2. Experimental set-up. Incubation cylinders including jellyfish treatment (right, N=3) and controls (left, N=3).**





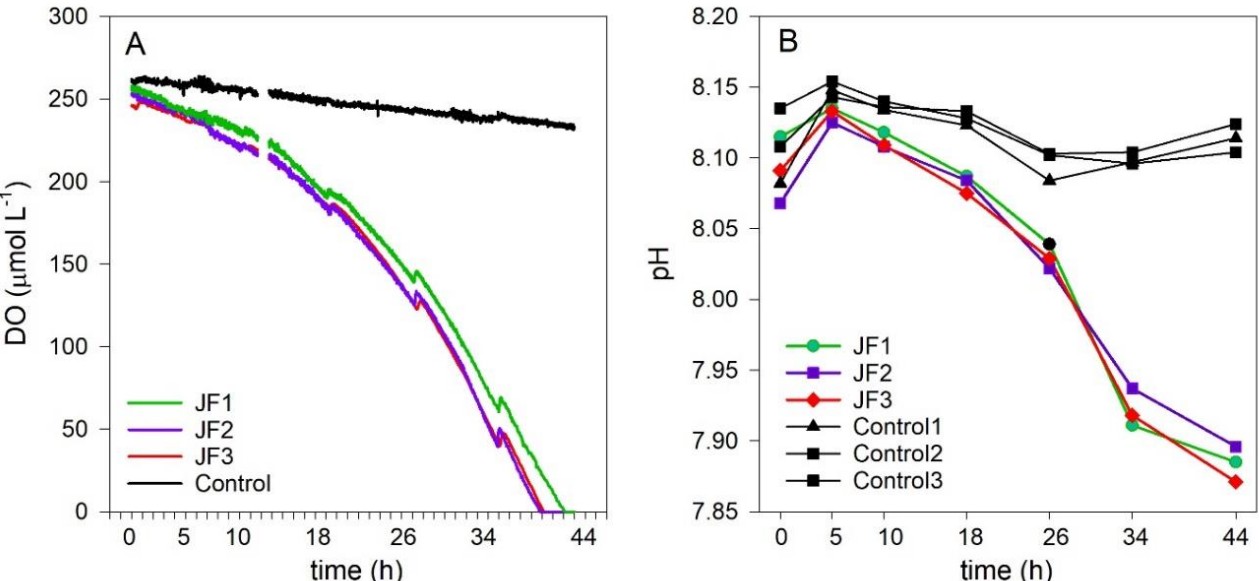

**Figure 3. The decomposition of the jellyfish *R. nomadica* leads to oxygen depletion and acidification in the seawater**
**overlying the sediment. A. Continuous dissolved oxygen (DO) record in the experimental cylinders enriched with**
**carcasses of the jellyfish *R. nomadica* (JF1-JF3) and in the controls. B. pH dynamics in the experimental cylinders,**
**including jellyfish and in the controls. N=3. The temperature was kept relatively constant at 27-28°C. The slight**
**increases in DO concentrations throughout the incubation period indicate water compensation during discrete**
**sampling events.**



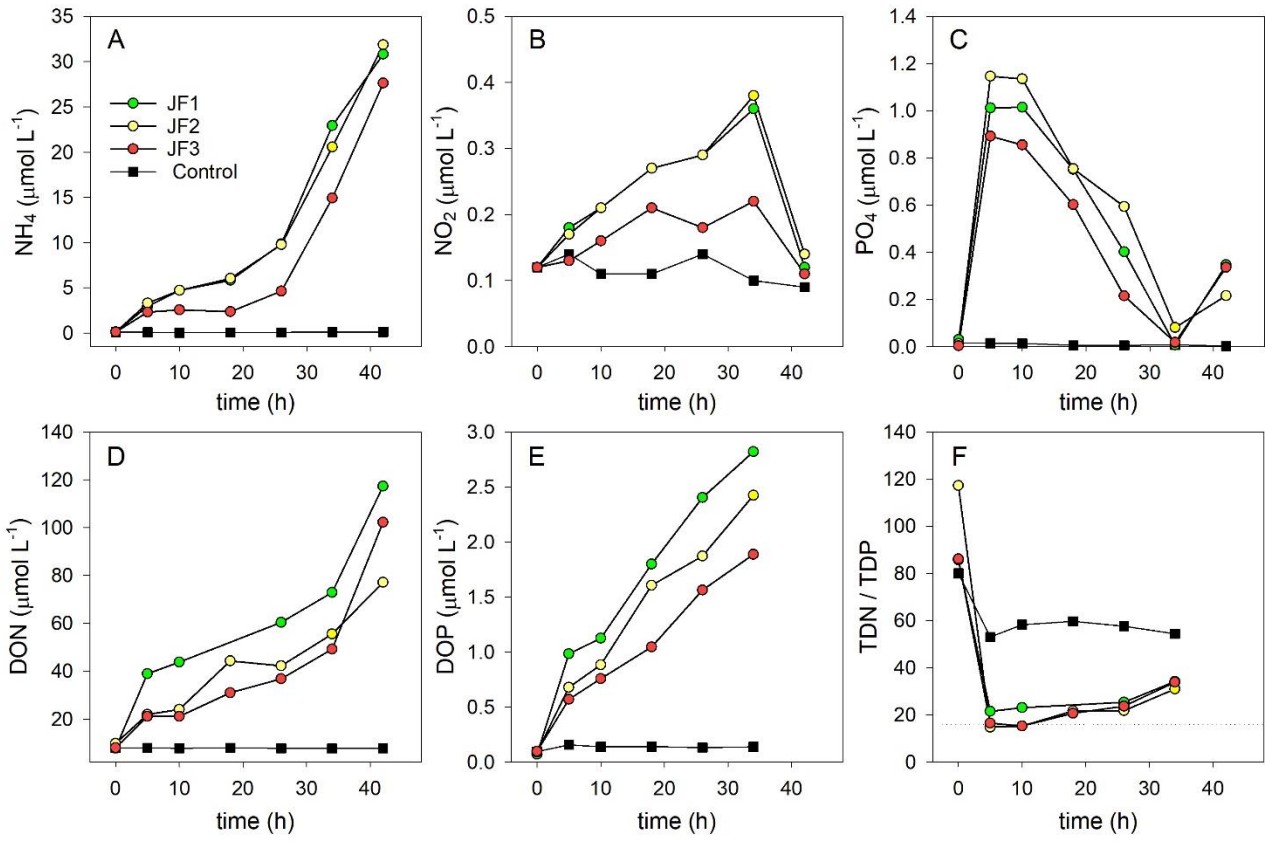


**Figure 4. Changes in the concentrations (µmol L$^{-1}$) of organic and inorganic nutrients in the experimental cylinders enriched with carcasses of the jellyfish *R. nomadica* and in the controls. A. ammonium. B. nitrite. C. orthophosphate. D. DON. E. DOP. F. TDN/TDP ratio. (N=3).**


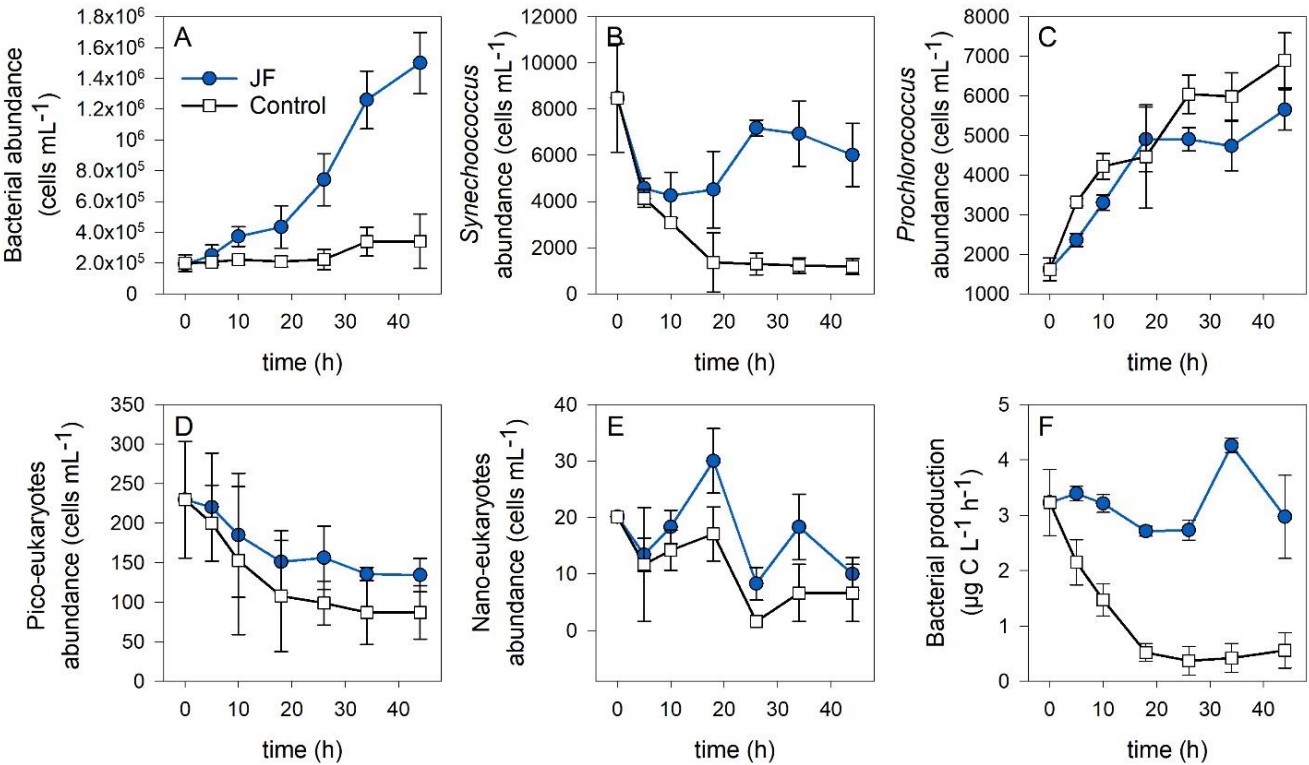

**Figure 5. Microbial abundance (cell·mL$^{-1}$) and production (µg C·L$^{-1}$·h$^{-1}$) in the jellyfish *R. nomadica* -enriched (blue) and control (black) experimental cylinders over the experimental period. A. total bacterial abundance. B. *Synechococcus*. C. *Prochlorococcus*. D. Pico-eukaryotes. E. Nano-eukaryotes. F. bacterial production. N=3, the error bars denote standard deviation.**



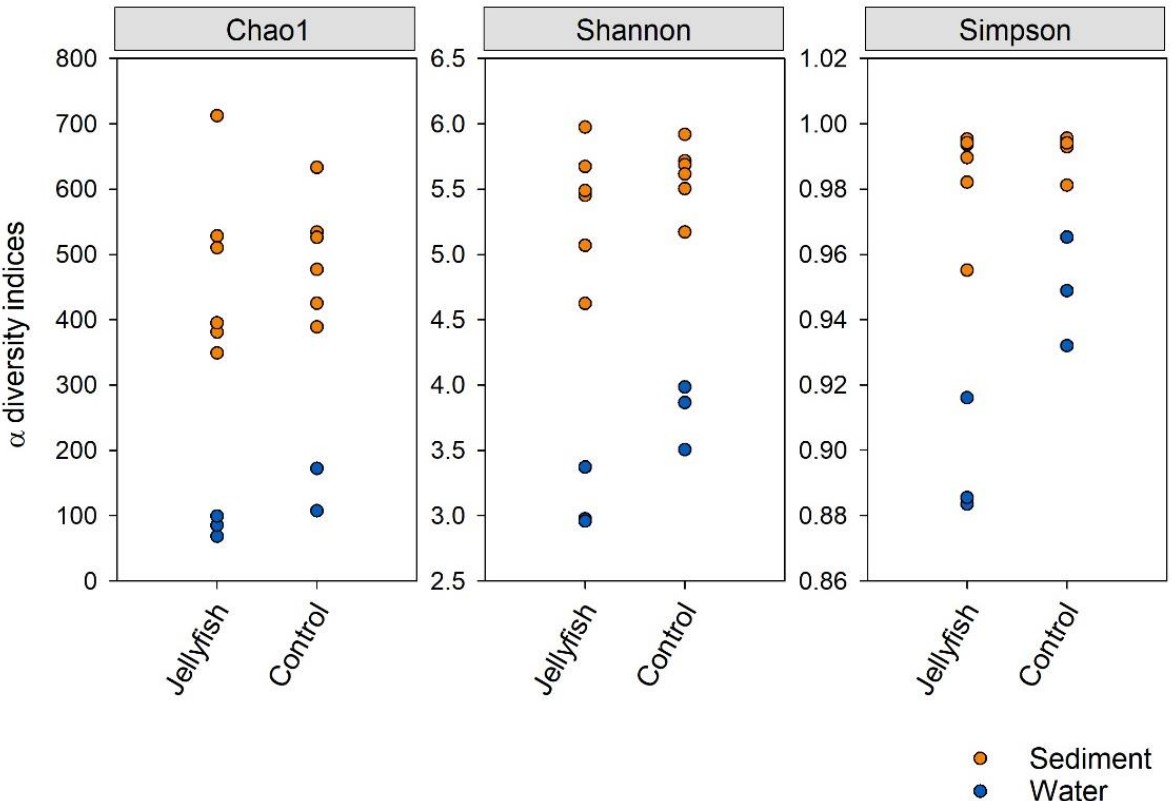

**Figure 6. Bacterial alpha diversity indices (Chao, Shannon, Simpson) in water and sediment samples from experimental cylinders enriched with carcasses of the jellyfish and in the controls (n=3).**



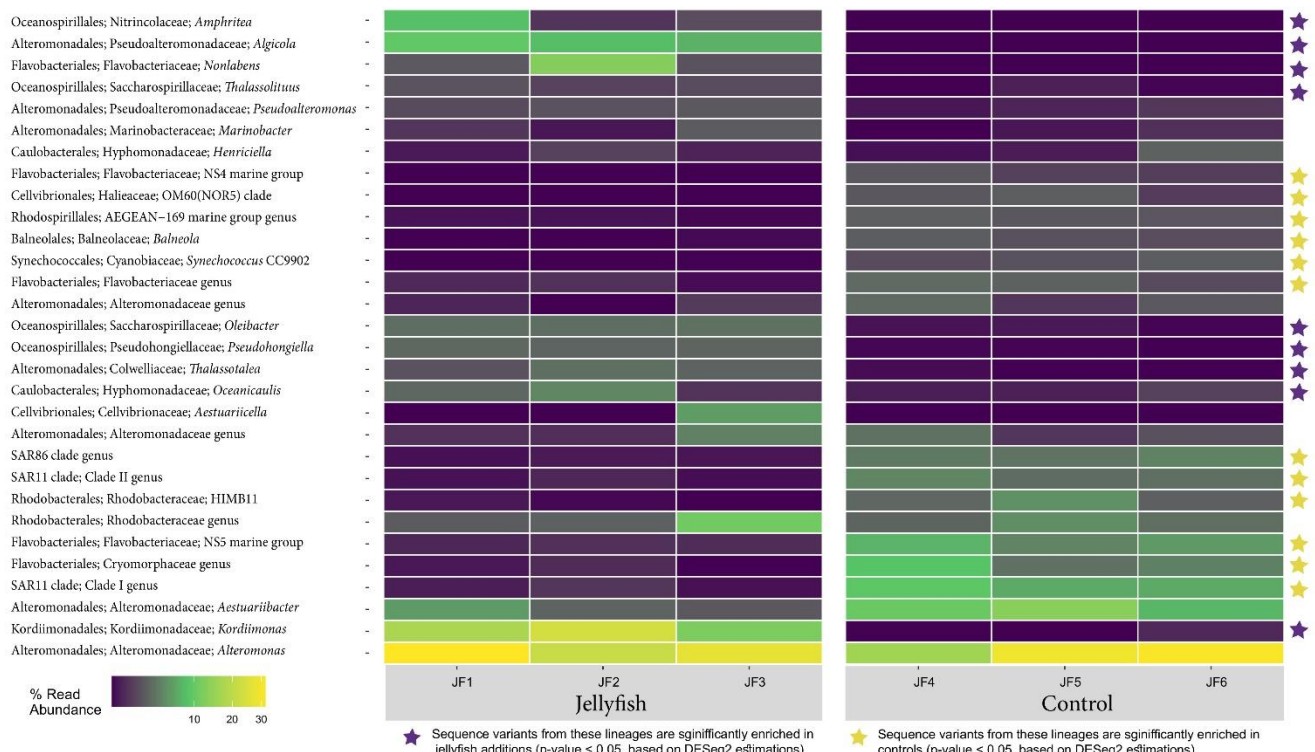

**Figure 7. Diversity of bacteria in the jellyfish-enriched and control experimental cylinder seawater. The 30 most abundant lineages are presented and organized by hierarchical clustering. Color scale denotes the relative abundance of reads (%). The star symbols on the right-side panel indicate lineages significantly more abundant in the jellyfish treatment (in purple) or the controls (in yellow) based on DESeq2 estimations.**





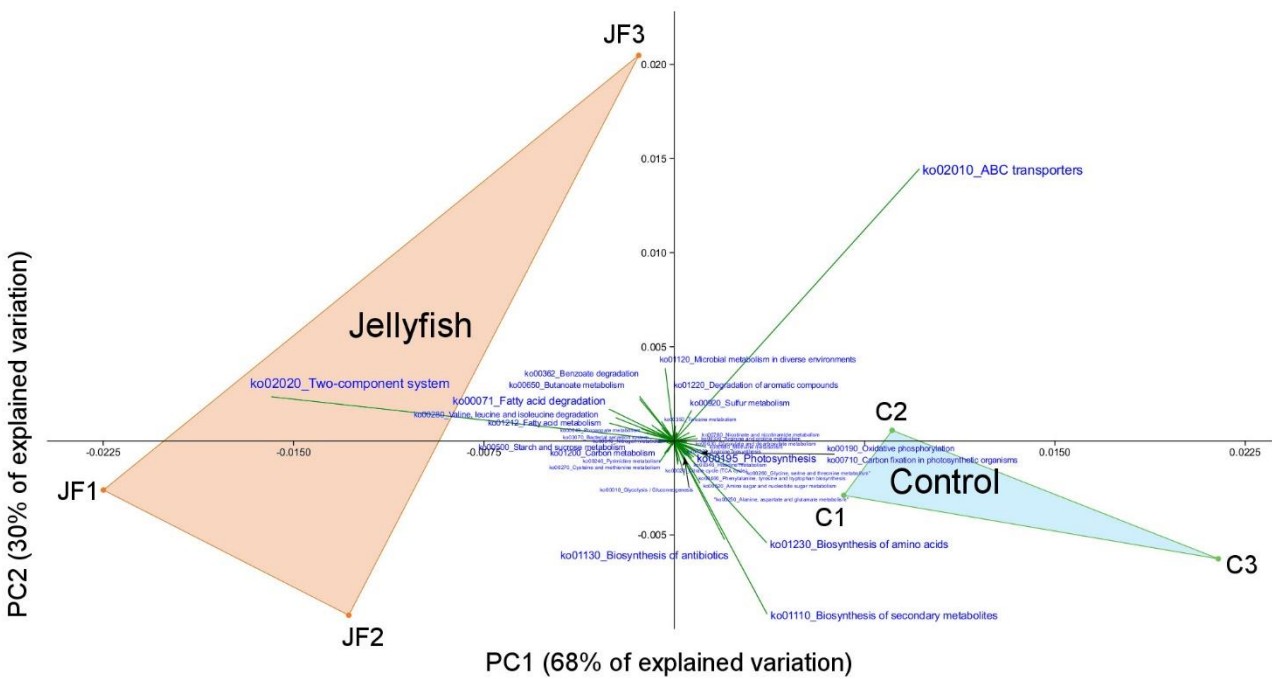


**Figure 8. Principle Component Analysis biplot of functional diversity based on taxonomy-based functional predictions using Tax4Fun2. JF = jellyfish samples; C = control samples. The vectors present KEGG pathways. The within-group similarity is 95% between the jellyfish treatments (red) and 97% between the controls (light blue).**




**Table 1: Daily oxygen consumption and nutrient release rates standardized to jellyfish (*R. nomadica*) biomass (µmol·g WW$^{-1}$·d$^{-1}$). The average wet weight of the whole jellyfish was 1.5±0.4 kg. N=3.**

|  | rate (µmol·g WW$^{-1}$·d$^{-1}$) | SD |
|---|---|---|
| DO | -17.9 | 0.3 |
| NH$_4$ | 2.0 | 0.2 |
| PO$_4$ | 0.6 | 0.1 |
| DON | 4.0 | 0.7 |
| DOP | 0.2 | 0.04 |

**Table 2: Calculated oxygen and nutrient fluxes in the seawater of jellyfish (*R. nomadica*) -enriched and control experimental cylinders. Positive flux represents water column enrichment (source), negative flux represent removal from the water column (sink). N=3. SD denotes standard deviation. N.A – not available.**

|  | Jellyfish (mmol m$^{-2}$ d$^{-1}$) | | Control (mmol m$^{-2}$ d$^{-1}$) | |
|---|---|---|---|---|
|  | Mean | SD | Mean | SD |
| DO | -56.9 | 1.0 | -6.7 | 0.3 |
| NH$_4$ (0-36 h) | 6.9 | 0.4 | $1 \cdot 10^{-2}$ | $8 \cdot 10^{-3}$ |
| PO$_4$ (0-5 h) | 1.9 | 0.2 | $-5 \cdot 10^{-3}$ | $1 \cdot 10^{-2}$ |
| DON | 12.7 | 2.4 | $-4 \cdot 10^{-2}$ | N.A. |
| DOP | 0.6 | 0.1 | $5 \cdot 10^{-3}$ | N.A. |






**Appendix A: Additional nutrient data**

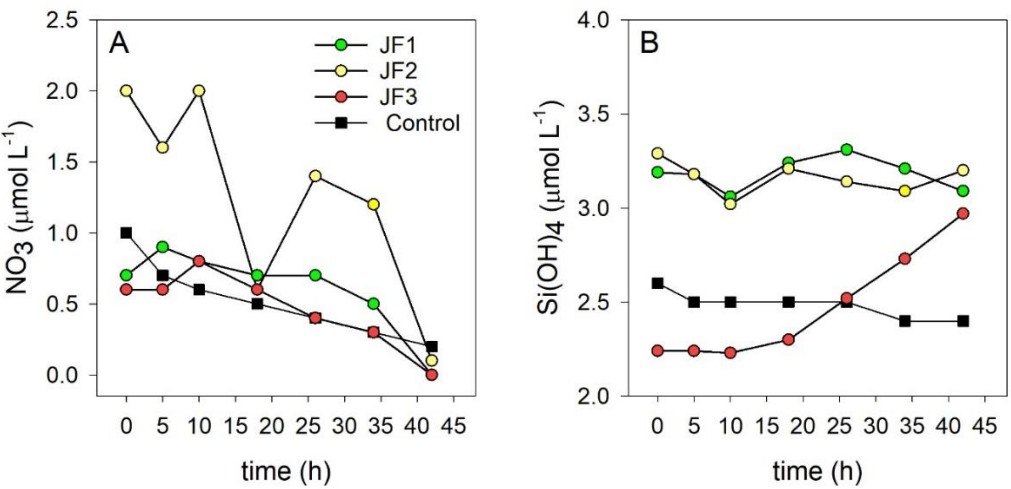

**Figure A1. Changes in the concentrations (µmol L$^{-1}$) of A. NO$_3$ and, B. Si(OH)$_4$ in the experimental cylinders enriched with carcasses of the jellyfish *R. nomadica* (JF1-JF3) and in the controls (N=3).**






**Appendix B: Nutrient--bacteria correlations**

**Table B1: Pearson correlation coefficients (*r*) between nutrient concentrations, bacterial abundance and production**
**rates. Averages of three replicates per time step were used (N=7). Significant correlations are marked in bold (p<0.05).**

|  | Bacterial abundance | Bacterial production |
|---|---|---|
| DO | **-0.995** | -0.211 |
| NH$_4$ | **0.979** | 0.236 |
| NO$_x$ | **-0.765** | 0.213 |
| PO$_4$ | -0.485 | -0.323 |
| Si(OH)$_4$ | **0.841** | 0.055 |
| DON | **0.944** | 0.038 |
| DOP | **0.912** | 0.164 |
| TDN | **0.954** | 0.355 |
| TDP | 0.632 | -0.027 |




# Appendix C: Bacterial and eukaryotic diversity in water and sediment samples

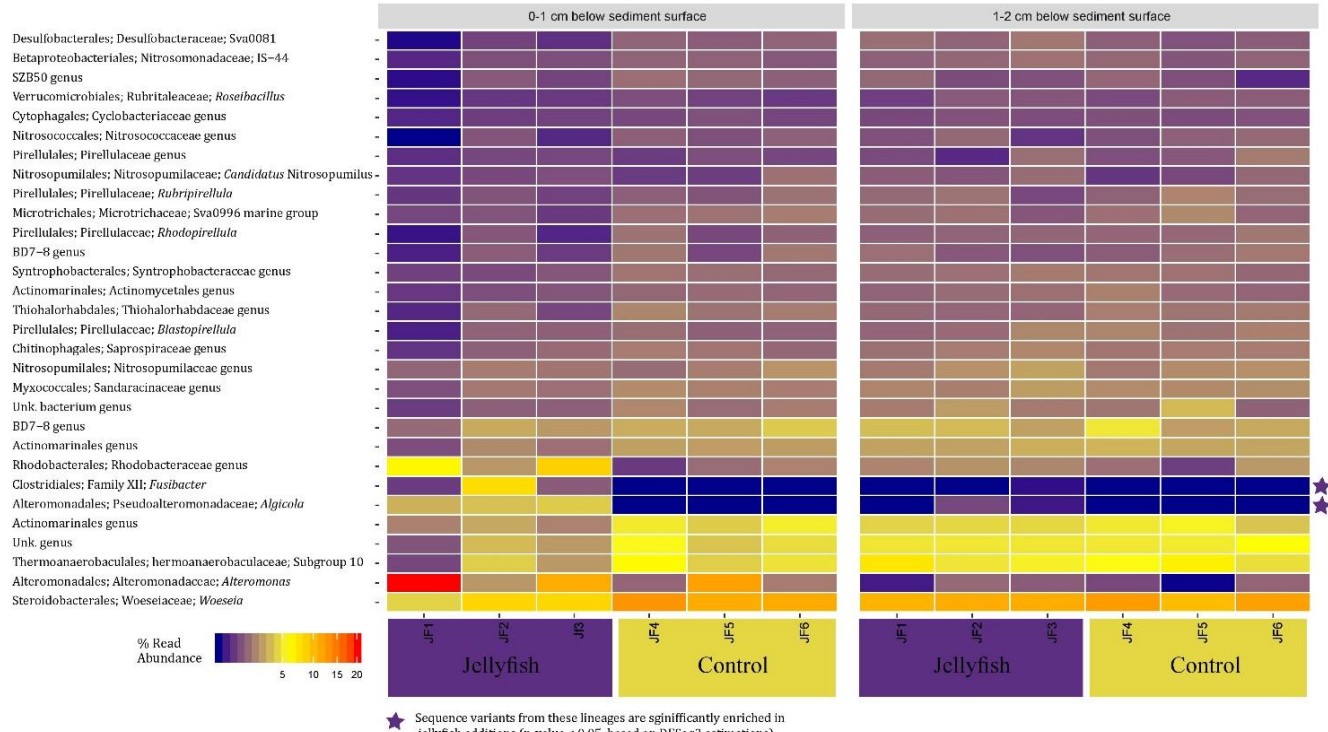

**Figure C1. Microbial diversity in sediment samples from the jellyfish-enriched and control experimental cylinders, from 0-1 cm (left) and 1-2 cm (right) depth layers. The 30 most abundant lineages are presented and organized by hierarchical clustering. Color scale denotes the relative abundance of reads (%). The star symbols on the right-side panel indicate lineages significantly more abundant in the jellyfish treatment based on DESeq2 estimations.**




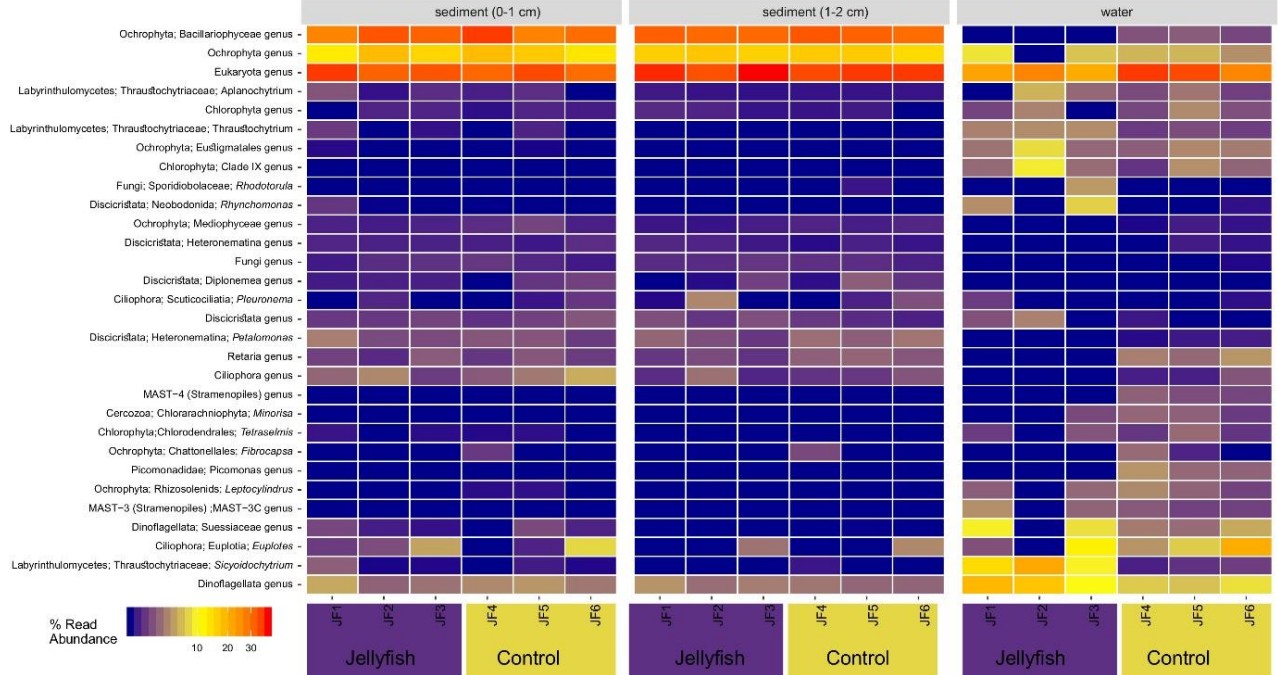

**Figure C2. Eukaryote diversity of seawater and sediment samples collected from jellyfish-enriched and control experimental cylinders. The 30 most abundant lineages are presented and organized by hierarchical clustering.. Color scale denotes the relative abundance of reads (%).**



**Figure C3. Rarefaction curves of observed 16S rRNA sequence variants retrieved from the seawater (upper graph) and sediment (lower graph) samples.**





**Figure C4. Rarefaction curves of observed 18S rRNA sequence variants retrieved from the seawater (upper graph) and sediment (lower graph) samples.**





**Appendix D: SIMPER analysis of main predicted functions based on KEGG orthologs**

730

**Table D1: Similarity Percentage (SIMPER) analysis indicating the main predicted functions characterizing the jellyfish and control communities (N=3). Av. Abund = Average abundance, Av. Sim = Average similarity, Sim/SD = similarity standard deviation, Contrib% = percent contribution, Cum.% = Cumulative contribution.**

**Jellyfish treatments**

Average similarity: 95.03

| KEGG ortholog | Predicted function | Av.Abund | Av.Sim | Sim/SD | Contrib% | Cum.% |
|---|---|---|---|---|---|---|
| ko01110 | Biosynthesis of secondary metabolites | 0.09 | 8.19 | 20.04 | 8.62 | 8.62 |
| ko01120 | Microbial metabolism in diverse environments | 0.08 | 7.42 | 458.33 | 7.81 | 16.43 |
| ko01130 | Biosynthesis of antibiotics | 0.07 | 6.87 | 33.95 | 7.23 | 23.66 |
| ko02020 | Two-component system | 0.07 | 6.41 | 25.99 | 6.74 | 30.4 |
| ko02010 | ABC transporters | 0.05 | 3.92 | 40.15 | 4.13 | 34.53 |
| ko01200 | Carbon metabolism | 0.04 | 3.87 | 44.28 | 4.08 | 38.61 |
| ko01230 | Biosynthesis of amino acids | 0.03 | 3.27 | 18.35 | 3.44 | 42.05 |

**Controls**

Average similarity: 97.47

| KEGG ortholog | Predicted function | Av.Abund | Av.Sim | Sim/SD | Contrib% | Cum.% |
|---|---|---|---|---|---|---|
| ko01110 | Biosynthesis of secondary metabolites | 0.09 | 9.08 | 91.13 | 9.32 | 9.32 |
| ko01120 | Microbial metabolism in diverse environments | 0.07 | 7.37 | 959.26 | 7.57 | 16.88 |
| ko01130 | Biosynthesis of antibiotics | 0.07 | 7.37 | 242.75 | 7.56 | 24.45 |





| KEGG ortholog | Predicted function | | | | | |
|---|---|---|---|---|---|---|
| ko02010 | ABC transporters | 0.06 | 5.48 | 31.5 | 5.62 | 30.07 |
| ko02020 | Two-component system | 0.05 | 4.63 | 6.87 | 4.75 | 34.81 |
| ko01230 | Biosynthesis of amino acids | 0.04 | 3.94 | 129 | 4.04 | 38.85 |
| ko01200 | Carbon metabolism | 0.04 | 3.88 | 1602.51 | 3.98 | 42.83 |

**Jellyfish treatments & controls**

Average dissimilarity: 7.01

| KEGG ortholog | Predicted function | Jellyfish Av.Abund | Control Av.Abund | Av.Diss | Diss/SD | Contrib% | Cum.% |
|---|---|---|---|---|---|---|---|
| ko02020 | Two-component system | 0.07 | 0.05 | 0.93 | 2 | 13.19 | 13.19 |
| ko02010 | ABC transporters | 0.05 | 0.06 | 0.7 | 3.51 | 10.02 | 23.21 |
| ko01110 | Biosynthesis of secondary metabolites | 0.09 | 0.09 | 0.35 | 1.35 | 5.05 | 28.27 |
| ko01230 | Biosynthesis of amino acids | 0.03 | 0.04 | 0.3 | 2.11 | 4.34 | 32.6 |
| ko01130 | Biosynthesis of antibiotics | 0.07 | 0.07 | 0.19 | 1.34 | 2.77 | 35.37 |
| ko00260 | Glycine, serine and threonine metabolism | 0.01 | 0.02 | 0.19 | 3.46 | 2.68 | 38.05 |
| ko00071 | Fatty acid degradation | 0.01 | 0.01 | 0.19 | 3.94 | 2.66 | 40.71 |

735