# Peer review of "The effects of decomposing invasive jellyfish on biogeochemical fluxes and microbial dynamics in an ultraoligotrophic sea"

_Biogeosciences, 2020_

## Referee Comment (RC1) · Anonymous Referee #1 · 21 Jul 2020

The manuscript describes changes in nutrients and microbial communities in a laboratory based jellyfish decomposition experiment. The manuscript is well written, the subject area is of interest and the particularly the biodiversity aspect is novel. However, the authors need to to take more account of the incubation system used more into account for the presentation and discussion of the data. Firstly, there is an evolution of processes during decomposition resulting from colonisation of the biomass, microbial growth dynamics and the sequential nature of the decomposition of particulate organic matter. Secondly in the discussion the limitations of the incubation method which resulted in large changes in conditions and in particular oxygen concentrations needs to be acknowledged and put into context of the smaller changes that would occur in situ.

[Figure]

Specific comments

In the abstract impacts on phytoplankton are mentioned, but there is no discussion of possible links between bloom decomposition and phytoplankton community structure and production in the introduction 33-45. As well as providing a food source to scavenging fauna, the presence of jellyfish carcasses on the sediment surface also simultaneously blocks oxygen transfer to the underlying sediment and stimulate anaerobic respiration processes, resulting in sediment reduction and accumulation of toxic sulphides (See cited Chelsky et al paper). These changes in sediment conditions result in migration or mortality of infauna, which are in turn a major influence on nutrient cycling (See for example Welsh 2000 Chemistry & Ecology 19, 321-342; Stief 2013 Biogeosciences 10, 2829-46 for reviews). These potential negative effects on benthic fauna and the indirect effect this has on nutrient cycling deserve a mention here, especially since they are again mentioned in the abstract.

65. This biomass addition is equivalent to approx. 3.5 kg per square metre. How realistic is this for a natural bloom collapse in the study area.

70-80. Were the cores incubated under light or dark conditions i.e. are there any effects of photoautotrophic activity on oxygen and nutrient concentrations

90-95. There are several issues with using this equation to calculate average fluxes over the entire incubation period as done in the results. Firstly, the equation assumes that the change in concentration is linear (consistent flux rate), but as the figure shows this is not true and fluxes rates evolve over time, as would be expected during decomposition (see cited decomposition studies), and in some cases reverse direction. At least in some cases, this impact could be minimised by calculating between time points, when conc changes would be closer to linear and changes between periods would show the evolution of flux rates over time. Secondly, fluxes are largely due to diffusion and diffusion rates depend on the concentration gradient between the sediment porewater and the overlying water. Therefore in a closed system like the one

used here, the changes in water column solute concentrations caused by the fluxes inhibit the rate of the flux that creates them by decreasing the concentration gradient between the sediment and water. This is especially true for oxygen where the water column conc falls to zero i.e. there is no oxygen consumption at the end of the experiment because there is no oxygen demand, but because there is no oxygen to supply the oxygen demand. Thirdly, as the extremely large change in water column oxygen concentration and therefore fluxes, aerobic processes become increasingly inhibited over time causing a shift to anaerobic processes, which would impact both nutrient dynamics and microbial community composition.

121-127. Presumably the 1.7 mL incubated refers to the seawater in the cores. However, it would be expected that the bulk of bacterial production would occur associated with the jellyfish tissues and the sediment in contact with these.

129-144. As above, this is not measuring the overall changes in populations, just those in the water column.

145-160. What statistical analyses were performed on the oxygen and nutrient data.

160-190. As above the effects of decomposition processes evolve over time due to colonisation processes, the sequential nature of decomposition e.g. PON decomposed to DON and DON to ammonium, shifting conditions and ultimately depletion of the biomass. This is shown by the non-linearity of the concentration changes that show that the production/consumption processes causing the fluxes are changing with in some cases the flux changing direction. Therefore, data need to be analysed in a manner that shows these shifting rates and the changing nutrient ratios they produce. It would also be useful to indicate what fraction of the C, N & P in the added biomass were actually mineralised over the course of the experiment. Especially as the data in the figure indicate that the decomposition rate had not even peaked by the end of the experiment, as ammonium production rates were still increasing at the end of the experiment. Indeed the highest rate of oxygen demand was at the end of the

experiment, despite low water column concentration present at this time.

204-208. There is no description of the sediment analyses in the methods section.

4.1. This section would be much improved by reanalysing the oxygen and nutrient flux with time. This would show how these evolved over time and how the composition of the TDN and TDP fluxes shifted over time. This would allow discussion of the decomposition process e.g. leaching versus decomposition, sequential mineralisation etc. Also some data on the proportion of particularly the N and P present in the biomass that was actually mineralised during the experiment would be useful, as it appears the decomposition process was only partially completed, so overall effects would be greater over longer time periods. Finally, some context needs to be given when making comparisons to the natural system e.g. how does the biomass density compare? How does a closed system with a 40 cm water column compare to in situ conditions with a large water column (greater oxygen available), which can be resupplied by water movements such as currents and exchange with the atmosphere i.e. potential in situ effects would be very, very much lower than those measured.

275-278. This N:P ratio is incorrect. It is not a %:% (weight:weight) ratio, it is an atom:atom (Mol:Mol) ratio. Therefore, the weights of N and P need to be divided by the atomic masses of N & P and the ratio of these compared.

317-324. Growth efficiency also depends on the type of respiration and decreases in the order of aerobic Approx. 0.5) > nitrate reduction > metal reductions > sulfate reduction (.0.2). Therefore, fixed production does not equal fixed rate of respiration as the type of respiration which is taking place shifts with oxygen conditions. Such changes would be even greater in jellyfish associated biofilms and in the surface sediments (See cited paper by Chelsky et al. 2016, which shows a shift to iron and sulfate reduction in the sediment in situ). The shift in your nitrate data from production (net nitrification) to consumption (net nitrate reduction), demonstrate this shift in dominance from aerobic to anaerobic processes in the benthos. Whereas, the water column effect in situ is

likely very, very different from thr changes that occurred in your cores.

317-361. Again the shifting conditions within the cores need to be discussed. As above there would be a shift from aerobic to anaerobic groups over time as conditions shifted. However, this would impact taxonomic diversity to a greater degree than functional diversity e.g. increased availability of protein would select for proteolytic bacteria, but oxygen availability would impact whether these were aerobic or anaerobic taxa of proteolytic bacteria.

---

## Referee Comment (RC2) · Anonymous Referee #2 · 29 Jul 2020

The paper of Guy-Haim et al. provides new information on the impact that the decomposition of jellyfish's carcasses can have on nutrients dynamics and on the bacteria living in sediments and in surrounding waters. The study focuses in particular on the jellyfish Rhopilema nomadica, a non indigenous species that has established in recent decades in some regions of the eastern Mediterranean, where swarms of this species are regularly reported with detrimental effects for different activities of high economical relevance. An experimental set-up is built to allow measuring nutrients and dissolved oxygen as well as assessing bacteria abundance, productivity and composition, throughout different phases of the carcasses' decomposition process. Results show that jellyfish degradation determines significant changes in nutrients supply, oxy-

gen concentration/ph and in the composition and abundance of bacteria living in the sediments and in the above water.

Overall the study addresses a highly relevant scientific question, providing a significant contribution towards a better understanding of the impact of jellyfish blooms on biogeochemical fluxes. Research outcomes here presented can be used to improve current ecosystem models, implementing the effects of jellyfish blooms, more specifically blooms of R. nomadica, on biogeochemical fluxes and on the first levels of the trophic web (i.e. bacterial communities).

The paper is quite comprehensive, though needs some revisions in the description of the methods and possibly in the presentation of some results. In particular session 2.6 should include more details on the numerical methods here adopted, as the reader is not necessarily familiar with the R routines indicated in the text and need to understand what has been done with the data. For instance it should be mentioned on which data set (supposedly 30 + 30 groups shown in Fig. 7 and fig. C1?) the diversity indices have been calculated and possibly why these three specific diversity indices (Chao, Shannon and Simpson) have been selected. Also, it should be indicated the dimension of the matrix (N metabolic functions/pathways X P observations) analysed by PCA, which should not include "rare" metabolic functions, i.e. lines with too many zeros, to prevent bias in the results of the analysis. Finally, Figure 8 should be redone using symbols and labels that would allow reading at least the key variables discussed in the text.

Further comments that should be also addressed before publication in Biogeoscience are listed here below:

- line 166: Table 2 should be cited instead of Table 1. -line 173: here it should be indicated that the NO3 concentration in JF2 is different from the other stations and possibly the reason for it should be discussed.

-Lines 302-307: this sentence is unclear and should be further revised. In particular it is not clear whether the chlorophyll maximum in late-spring summer is a recurrent

event that does usually follow records of jellyfish blooms. Unless the two events can be chronologically connected, the sentence here drafted should be changed or deleted; -Line 314: in the first and second parentheses Synechococcus and Prochlorococcus should be respectively indicated (in other words, the two parentheses have been inverted). - Lines 324-325: I suggest to revise the text along the following lines: "In the shallow waters of the EMS the peak of bacterial production observed in summer is possibly associated with the swarms of R. nomadica, which are frequently (regularly?) observed in this season" -Lines 363-365: this sentence needs further revision, as the study does not really measure decomposition dynamics in the Mediterranean, which would imply measurements done in situ. The study does rather measure nutrients and dissolved oxygen released by remineralisation of R. nomadica carcasses and the potential impact of this on the bacterial community.

---

## Author Comment (AC1) · 10 Aug 2020

Reviewer comments are in italics, answers follow within a text box in a dark blue font. The line numbers are compatible with the revised manuscript.

**REVIEWER #1: Anonymous**

**General comments:**

The manuscript describes changes in nutrients and microbial communities in a laboratory-based jellyfish decomposition experiment. The manuscript is well written, the subject area is of interest and particularly the biodiversity aspect is novel.

[**R1.1**] The authors need to take more account of the incubation system used for the presentation and discussion of the data. Firstly, there is an evolution of processes during decomposition resulting from colonisation of the biomass, microbial growth dynamics and the sequential nature of the decomposition of particulate organic matter.

> Our study was aimed at measuring fluxes at the sediment-water interface following jellyfish (specifically *R. nomadica*) decomposition. To do that correctly, we had to use the core incubation technique, which limited the temporal resolution of our study. We define this aim and acknowledge the method limitations in lines 77-80: "Nutrient fluxes were measured using the whole core incubation technique previously described by Denis et al. (2001). Although restricting this study for testing short term responses, this method follows the best practices for measuring oxygen and nutrient fluxes and dynamics at the sediment-water interface (Glud, 2008; Hammond et al., 2004; Pratihary et al., 2014; Skoog and Arias-Esquivel, 2009)".

[**R1.2**] Secondly, in the discussion the limitations of the incubation method which resulted in large changes in conditions and in particular oxygen concentrations needs to be acknowledged and put into context of the smaller changes that would occur in situ.

> We accept the reviewer's recommendation. In the revised manuscript, we added the following text to the Discussion (lines 256-257): "Here we found that the decomposition of the invasive jellyfish *Rhopilema nomadica* triggered deoxygenation of the seawater overlying the sediment to hypoxic and eventually anoxic levels, although the complete dissipation of oxygen is likely due to the experimental conditions". Nevertheless, we have recently performed a large-scale experiment (in a climate-change context), in a flow-through mesocosm system with high flux rate using realistic concentrations of *R. nomadica* carcasses, and measured low oxygen (hypoxic) levels in the water column in the first 24 hours of exposure. These results will be shown in a different separated publication focused on ocean warming.

**Specific comments:**

[**R1.3**] In the abstract, impacts on phytoplankton are mentioned, but there is no discussion of possible links between bloom decomposition and phytoplankton community structure and production in the introduction 33-45.

> In the revised manuscript, we have added to the Introduction the following text (lines 38-41): "Both in the water column and on the sediment, jelly-falls undergo bacterial decomposition, directly affecting nutrient cycling (Qu et al., 2015; West et al., 2008), potentially altering plankton community composition (Xiao et al., 2019) and stimulating algal blooms (Møller and Riisgård, 2007)".
>
> The link between decomposition and phytoplankton community structure is further discussed in lines 308-321.

[**R1.4**] As well as providing a food source to scavenging fauna, the presence of jellyfish carcasses on the sediment surface also simultaneously blocks oxygen transfer to the underlying sediment and stimulate anaerobic respiration processes, resulting in sediment reduction and accumulation of toxic sulphides (See cited Chelsky et al paper). These changes in sediment conditions result in migration or mortality of infauna, which are in turn a major influence on nutrient cycling (See for example Welsh 2000 Chemistry & Ecology 19, 321-342; Stief 2013 Biogeosciences 10, 2829-46 for reviews). These potential negative effects on benthic fauna and the indirect effect this has on nutrient cycling deserve a mention here, especially since they are again mentioned in the abstract.

> Following the reviewer's suggestion, we have added the following text to the Introduction (lines 38-41): "Both in the water column and on the sediment, jelly-falls undergo bacterial decomposition, directly affecting nutrient cycling (Qu et al., 2015; West et al., 2008). Changes in the sediment conditions may result in migration or mortality of infauna (Chelsky et al., 2016), which in turn affect indirectly nutrient cycling (Stief, 2013; Welsh, 2003)".
>
> Additionally, the potential negative effects of jellyfish decomposition on benthic fauna are mentioned throughout the Discussion: deoxygenation and acidification (line 274-279), ammonium toxicity effect (lines 279-280), and dissolved sulfides (lines 308-309).

[**R1.5**] L65. This biomass addition is equivalent to approx. 3.5 kg per square metre. How realistic is this for a natural bloom collapse in the study area?

> Our biomass estimation is realistic based on the published data on the density of *R. nomadica* in blooms in the EMS, as well as personal observations (please see an example below). The jellyfish densities are discussed in lines 302-307: "Reported densities of *R. nomadica* aggregations from the EMS are $1.6 \cdot 10^5$ km$^{-2}$ in the Israeli coast (Lotan et al., 1992; Lotan et al., 1994), $1 \cdot 10^6$ km$^{-2}$ in the Lebanese coast (Lakkis and Zeidane, 1991), and $9 \cdot 10^5$ km$^{-2}$ in the Mediterranean Egyptian coast (Madkour et al., 2019). The average wet weight of *R. nomadica* changes seasonally, 1340 ±953 g ind$^{-1}$ during summer and 2450 ±1854 g ind$^{-1}$ during winter (N=40, T.G.-H. unpublished data), yielding ca. 1.3 kt km$^{-2}$". Accordingly, the densities of *R. nomadica* blooms in the EMS are 0.2-1 ind m$^{-2}$. Therefore, our biomass estimation (25 g 78.5 cm$^{-2}$ = 3.2 kg m$^{-2}$) falls at the high limits within the realistic range. In particularly, jelly-falls of

of 1-5 ind m$^{-2}$ on the sediment, depending on substrate topography (see below photos), thus our biomass concentration in the experiment may have been an underestimation. We plan to present *in-situ* measurements of such jelly-falls in a future publication.

[Figure]

Jelly-falls of *Rhopilema nomadica* and *Rhizostoma pulmo* in Dor, Israel (15-m depth). Photos: courtesy of H. Nativ (University of Haifa).

[**R1.6**] L70-80. Were the cores incubated under light or dark conditions i.e. are there any effects of photoautotrophic activity on oxygen and nutrient concentrations.

> The cores were incubated under PAR= 100 µmol photons·m$^{-2}$·s$^{-1}$ with a photoperiodicity of 14:10 (L:D). This information was added to the Methods section (lines 83-84).

[**R1.7**] L90-95. There are several issues with using this equation to calculate average fluxes over the entire incubation period as done in the results. Firstly, the equation assumes that the change in concentration is linear (consistent flux rate), but as the figure shows this is not true and fluxes rates evolve over time, as would be expected during decomposition (see cited decomposition studies), and in some cases reverse direction. At least in some cases, this impact could be minimised by calculating between time points, when conc changes would be closer to linear and changes between periods would show the evolution of flux rates over time.

> We thank the reviewer for this important comment. In the revised manuscript, we address the non-linearity of fluxes by analyzing them over time, i.e., applying several linear phases (lines 175-199, 288-291). We also address the changes in the direction of the fluxes, which were evident in NO$_x$ and PO$_4$. To calculate the diel fluxes (Tables 1,2), we have integrated the changes over time and indicated the time span used for calculation.

[**R1.8**] Secondly, fluxes are largely due to diffusion and diffusion rates depend on the concentration gradient between the sediment porewater and the overlying water. Therefore, in a closed system like the one used here, the changes in water column solute concentrations caused by

the fluxes inhibit the rate of the flux that creates them by decreasing the concentration gradient between the sediment and water. This is especially true for oxygen where the water column conc falls to zero i.e. there is no oxygen consumption at the end of the experiment because there is no oxygen demand, but because there is no oxygen to supply the oxygen demand.

> In our experiment, the controls (N=3) showed no change in nutrient concentrations (Fig. 4). Therefore, although we have not measured pore water chemistry, it can be reasonably assumed that diffusive flux is negligible. The higher oxygen flux rate at the end of the experiment (under low oxygen concentration) is not due to diffusive flux but rather due to heterotrophic microbial activity.

[**R1.9**] Thirdly, as the extremely large change in water column oxygen concentration and therefore fluxes, aerobic processes become increasingly inhibited over time causing a shift to anaerobic processes, which would impact both nutrient dynamics and microbial community composition.

> We acknowledge that over time (in our experiment, after >26 hrs) oxygen levels were reduced to hypoxic levels (<4 mg $L^{-1}$), impacting both nutrient dynamics and microbial community composition. Nevertheless, nutrient flux and bacterial abundance and production within the first 24 hours of exposure show large changes which we focus on in the discussion.

[**R1.10**] L121-127. Presumably the 1.7 mL incubated refers to the seawater in the cores. However, it would be expected that the bulk of bacterial production would occur associated with the jellyfish tissues and the sediment in contact with these.

> The 1.7 ml water samples for bacterial productivity incubations as well as the 1.6 ml samples for flow cytometry were drawn at each time point from above the jellyfish/sediment (see Fig. 2). Both bacterial abundance and production were significantly higher in the jellyfish treatments than in the controls (at respective depth), suggesting that not only the jellyfish tissue and the below sediment but also the overlying waters are affected by increased bacterial abundance and production. Studying the jellyfish epi-biome microbial dynamics is out of the scope of the present study.

[**R1.11**] L129-144. As above, this is not measuring the overall changes in populations, just those in the water column.

> See the response to R1.10. The samples were taken from the overlying waters and represent the communities at the sediment-water interface.

[**R1.12**] L145-160. What statistical analyses were performed on the oxygen and nutrient data?

> Oxygen and nutrient data were correlated with bacterial abundance and bacterial production using Pearson correlation using R package Hmisc (Harrell, 2004). See line 163 and Table B1.

[**R1.13**] L160-190. As above the effects of decomposition processes evolve over time due to colonisation processes, the sequential nature of decomposition e.g. PON decomposed to DON and DON to ammonium, shifting conditions and ultimately depletion of the biomass. This is shown by

the non-linearity of the concentration changes that show that the production/consumption processes causing the fluxes are changing with in some cases the flux changing direction. Therefore, data need to be analysed in a manner that shows these shifting rates and the changing nutrient ratios they produce. It would also be useful to indicate what fraction of the C, N & P in the added biomass were actually mineralised over the course of the experiment. Especially as the data in the figure indicate that the decomposition rate had not even peaked by the end of the experiment, as ammonium production rates were still increasing at the end of the experiment. Indeed the highest rate of oxygen demand was at the end of the experiment, despite low water column concentration present at this time.

> We thank the reviewer for this important comment. Indeed the non-linearity of the concentration changes indicate a sequential nature of decomposition, likely due colonization and POM breakdown. In the revised manuscript (lines 175-199, 288-291), we have addressed the non-linearity of fluxes by analyzing them over time, i.e., applying several linear phases. We also address the changes in the direction of the fluxes, which were evident in $NO_x$ and $PO_4$. To calculate the diel fluxes (Tables 1,2), we have integrated the changes over time and indicated the time span used for calculation.

[**R1.14**] There is no description of the sediment analyses in the methods section.

> This information was already included in the initial manuscript in the Methods section (lines 135-137): "250 mg from 0-1 and 1-2 cm sediment sections were transferred into the extraction tube. DNA was extracted from water and sediment using the DNeasy PowerSoil Kit (Qiagen, California, USA), using the manufacturer's protocol that included a FastPrep-24™ (MPBIO, Ohio, USA) bead-beating step (2x40 sec at 5.5 m/s, with a 5 min interval)".

[**R1.15**] 4.1. This section would be much improved by reanalysing the oxygen and nutrient flux with time. This would show how these evolved over time and how the composition of the TDN and TDP fluxes shifted over time. This would allow discussion of the decomposition process e.g. leaching versus decomposition, sequential mineralisation etc. Also some data on the proportion of particularly the N and P present in the biomass that was actually mineralised during the experiment would be useful, as it appears the decomposition process was only partially completed, so overall effects would be greater over longer time periods. Finally, some context needs to be given when making comparisons to the natural system e.g. how does the biomass density compare? How does a closed system with a 40 cm water column compare to in situ conditions with a large water column, which can be resupplied by water movements such as currents and exchange with the atmosphere i.e. potential in situ effects would be very, very much lower than those measured.

> This is a summary of former comments made by the reviewer. See responses to R1.1-R1.14.

[**R1.16**] L275-278. This N:P ratio is incorrect. It is not a %:% (weight:weight) ratio, it is an atom:atom (Mol:Mol) ratio. Therefore, the weights of N and P need to be divided by the atomic masses of N & P and the ratio of these compared.

> We thank the reviewer for this comment. The study N:P ratios throughout the manuscript are presented correctly as mol:mol ratios. However, the N:P ratio derived from Lucas et al. (2011) was incorrectly calculated from %:%. In the revised manuscript, this ratio was corrected to mol:mol (lines 296-297): "Elemental body composition of scyphozoan jellyfish, in general, is 2.48 N %DW (dry weight) and 0.22 P %DW, hence an N:P ratio of 25:1 (Lucas et al., 2011)".

[**R1.17**] L317-324. Growth efficiency also depends on the type of respiration and decreases in the order of aerobic Approx. 0.5) > nitrate reduction > metal reductions > sulfate reduction (.0.2). Therefore, fixed production does not equal fixed rate of respiration as the type of respiration, which is taking place shifts with oxygen conditions. Such changes would be even greater in jellyfish associated biofilms and in the surface sediments (See cited paper by Chelsky et al. 2016, which shows a shift to iron and sulfate reduction in the sediment in situ). The shift in your nitrate data from production (net nitrification) to consumption (net nitrate reduction), demonstrate this shift in dominance from aerobic to anaerobic processes in the benthos. Whereas, the water column effect in situ is likely very, very different from the changes that occurred in your cores.

We agree. Following the reviewer's suggestion, we have added to the revised manuscript (lines 291-294): "The shift from nitrate production to nitrate consumption 36 hours from the onset of the experiment likely reflects the shift from aerobic to anaerobic processes due to the low, hypoxic (and eventually anoxic) levels and may be regarded as an experimental artefact, although such changes were previously showed in surface sediments (Chelsky et al., 2016)".

---

## Author Comment (AC2) · 10 Aug 2020

Reviewer comments are in italics, answers follow within a text box in a dark blue font. The line numbers are compatible with the revised manuscript.

**REVIEWER #2: Anonymous**

**General comments:**

The paper of Guy-Haim et al. provides new information on the impact that the decomposition of jellyfish's carcasses can have on nutrients dynamics and on the bacteria living in sediments and in surrounding waters. The study focuses in particular on the jellyfish *Rhopilema nomadica*, a non-indigenous species that has established in recent decades in some regions of the eastern Mediterranean, where swarms of this species are regularly reported with detrimental effects for different activities of high economical relevance. An experimental set-up is built to allow measuring nutrients and dissolved oxygen as well as assessing bacteria abundance, productivity and composition, throughout different phases of the carcasses' decomposition process. Results show that jellyfish degradation determines significant changes in nutrients supply, oxygen concentration/pH and in the composition and abundance of bacteria living in the sediments and in the above water.

Overall, the study addresses a highly relevant scientific question, providing a significant contribution towards a better understanding of the impact of jellyfish blooms on biogeochemical fluxes. Research outcomes here presented can be used to improve current ecosystem models, implementing the effects of jellyfish blooms, more specifically blooms of *R. nomadica*, on biogeochemical fluxes and on the first levels of the trophic web (i.e. bacterial communities).

[**R2.1**] The paper is quite comprehensive, though needs some revisions in the description of the methods and possibly in the presentation of some results. In particular, session 2.6 should include more details on the numerical methods here adopted, as the reader is not necessarily familiar with the R routines indicated in the text and need to understand what has been done with the data.

> Following the reviewer's suggestion, in the revised manuscript we have detailed all abbreviations used for statistical methods (lines 156-165). The full details of the statistical and bioinformatics methods can be found in the cited references, and are customary in studies of microbial community diversity using amplicon sequences (reviewed in Knight et al., 2018, Prodan et al., 2020).
>
> Knight, R., Vrbanac, A., Taylor, B.C., Aksenov, A., Callewaert, C., Debelius, J., Gonzalez, A., Kosciolek, T., McCall, L.I., McDonald, D. and Melnik, A.V., 2018. Best practices for analysing microbiomes. *Nature Reviews Microbiology*, *16*(7), pp.410-422.
>
> Prodan, A., Tremaroli, V., Brolin, H., Zwinderman, A.H., Nieuwdorp, M. and Levin, E., 2020. Comparing bioinformatic pipelines for microbial 16S rRNA amplicon sequencing. *Plos one*, *15*(1), p.e0227434.

[**R2.2**] For instance, it should be mentioned on which data set (supposedly 30 + 30 groups shown in Fig. 7 and fig. C1?) the diversity indices have been calculated and possibly why these three specific diversity indices (Chao, Shannon and Simpson) have been selected.

> The dataset on which Fig. 7 and Fig. C1 are based on is the 16S amplicon sequences obtained by Illumina high-throughput sequencing (see Materials and Methods section 2.5), which was analyzed according to the described pipeline (see Materials and Methods section 2.6). The dataset was deposited in NCBI GenBank, in the Sequence Read Archive (SRA): BioProject PRJNA626084 (see section Data Availability).
>
> The alpha diversity indices used in our study (Chao1, Shannon and Simpson) are the most common indices used in microbial diversity research to compare the diversity among samples and between treatments with controls. Chao1 is an abundance-based estimator of species richness. Simpson Index is an estimator of species richness and species evenness, with more weight on species evenness; whereas Shannon Index is estimator of species richness and species evenness, with more weight on species richness. See the following review:
>
> Kim, B.R., Shin, J., Guevarra, R., Lee, J.H., Kim, D.W., Seol, K.H., Lee, J.H., Kim, H.B. and Isaacson, R.E., 2017. Deciphering diversity indices for a better understanding of microbial communities. *J Microbiol Biotechnol*, *27*(12), pp.2089-2093.

[**R2.3**] Also, it should be indicated the dimension of the matrix (N metabolic functions/pathways X P observations) analysed by PCA, which should not include "rare" metabolic functions, i.e. lines with too many zeros, to prevent bias in the results of the analysis.

> The PCA matrix included 324 KEGG orthologs (KOs) across all samples (N=3 jellyfish-treatments and N=3 controls), after reducing rare KOs (appearing in only one replicate), to avoid zero-inflated dimensionality. This information was added to the revised manuscript (lines 241-244, Fig. 8 caption).

[**R2.4**] Finally, Figure 8 should be redone using symbols and labels that would allow reading at least the key variables discussed in the text.

[Figure]

We have modified Figure 8 to include larger labels (detailed in caption) for a better readability.

[**R2.5**] line 166: Table 2 should be cited instead of Table 1.

> Table 1 details the diel oxygen and nutrient fluxes standardized per jellyfish biomass ($\mu$mol·g WW$^{-1}$·d$^{-1}$) whereas Table 2 presents the fluxes at the sediment-water interface as measured in the experimental chambers standardized to square meter (mmol m$^{-2}$ d$^{-1}$). In line 166 (lines 171-172 in the revised text), we discuss the fluxes at the sediment-water interface as measured in the jellyfish treatment chambers versus the controls. Thereby, Table 1 is referred.

[**R2.6**] line 173: here it should be indicated that the NO$_3$ concentration in JF2 is different from the other stations and possibly the reason for it should be discussed.

> The following text was added (lines 183-185): "One of the jellyfish treatments (JF2) showed higher (2-fold) concentrations of NO$_3$ throughout the experiment, likely due to a different initial NO$_3$ content derived from the mixture of jellyfish tissue, as some parts have shown to include higher concentrations of dissolved nitrogen (MacKenzie et al., 2017). Nevertheless, this has not affected the overall nutrient fluxes nor triggered different responses to the microbial communities (thus, the same direction and strength of responses were observed in all jellyfish addition treatments)".

[**R2.7**] Lines 302-307: this sentence is unclear and should be further revised. In particular, it is not clear whether the chlorophyll maximum in late-spring summer is a recurrent event that does usually follow records of jellyfish blooms. Unless the two events can be chronologically connected, the sentence here drafted should be changed or deleted.

> In this section, we describe the discrepancy between the high Chl-*a* concentrations in the water column of the EMS coastal waters during winter, to the chlorophyll maximum in the sediment in late-spring summer, which was previously explained by spring bloom of benthic producers. We suggest that the summer decomposition of *R. nomadica* blooms may also contribute to the high summer concentrations measured in the sediment throughout the leaching of limiting nutrients to phytoplankton (namely N and P). For better clarification, in the revised manuscript we have emphasized the differences between the water column and sediment (lines 317-319).

[**R2.8**] Line 314: in the first and second parentheses Synechococcus and Prochlorococcus should be respectively indicated (in other words, the two parentheses have been inverted).

> Corrected: "Autotrophic cyanobacteria, on the other hand, decreased (*Synechococcus*), or increased to a lower level than the unamended control (*Prochlorococcus*)" likely due to deoxygenation (Bagby and Chisholm, 2015) or out-competition…" (lines 327-329).

[**R2.9**] Lines 324-325: I suggest to revise the text along the following lines: "In the shallow waters of the EMS the peak of bacterial production observed in summer is possibly associated with the swarms of *R. nomadica*, which are frequently (regularly?) observed in this season"

> The swarms of *R. nomadica* in the EMS are observed semi-annually, during both winter and summer, coinciding with the bacterial production peaks in the EMS shallow waters. However, we would like to be more careful with our statement, and therefore we prefer using "potentially contributing" than "associated with" that may suggest causality.

[**R2.10**] Lines 363-365: this sentence needs further revision, as the study does not really measure decomposition dynamics in the Mediterranean, which would imply measurements done in situ. The study does rather measure nutrients and dissolved oxygen released by remineralisation of *R. nomadica* carcasses and the potential impact of this on the bacterial community.

> We revised the sentence following the reviewer's suggestion: "Our study examined, for the first time, the decomposition effects of the bloom-forming invasive jellyfish *R. nomadica* on the oxygen and nutrient fluxes and microbial communities at the sediment-water interface" (lines 377-378).
>
> Using this experimental setup is the best practice for the study of fluxes at the sediment-water interface (Denis et al., 2001; Glud, 2008; Hammond et al., 2004; Pratihary et al., 2014; Skoog and Arias-Esquivel, 2009). Yet, we agree that *in-situ* measurements are necessary for assessing post-bloom dynamics. In the Conclusions section, we included a paragraph (lines 387-391) on necessary future research. Indeed, we are currently running thermal large-scale mesocosms and *in-situ* research that we aim at summarizing in future publications.